# New Insight into the Related Candidate Genes and Molecular Regulatory Mechanisms of Waterlogging Tolerance in Tree Peony *Paeonia ostii*

**DOI:** 10.3390/plants13233324

**Published:** 2024-11-27

**Authors:** Minghui Zhou, Xiang Liu, Jiayan Zhao, Feng Jiang, Weitao Li, Xu Yan, Yonghong Hu, Junhui Yuan

**Affiliations:** 1School of Ecological Technology and Engineering, Shanghai Institute of Technology, No. 100 Haiqun Rd., Fengxian District, Shanghai 201416, China; zmh1882022@126.com (M.Z.); liuxiang@sit.edu.cn (X.L.); zhaojiay0111@126.com (J.Z.); ujiangfeng@126.com (F.J.); 2Shanghai Key Laboratory of Plant Functional Genomics and Resources, Shanghai Chenshan Botanical Garden, No. 3888 Chenhua Rd., Songjiang District, Shanghai 201602, China; starlinanul@163.com (W.L.); ianmooneyx@gmail.com (X.Y.)

**Keywords:** *Paeonia ostii*, root system, waterlogging tolerance, transcriptome analysis

## Abstract

Research on the waterlogging tolerance mechanisms of *Paeonia ostii* helps us to further understand these mechanisms in the root system and enhance its root bark and oil yields in southern China. In this study, root morphological identification, the statistics of nine physiological and biochemical indicators, and a comparative transcriptome analysis were used to investigate the waterlogging tolerance mechanism in this plant. As flooding continued, the roots’ vigor dramatically declined from 6 to 168 h of waterlogging, the root number was extremely reduced by up to 95%, and the number of roots was not restored after 96 h of recovery. Seven of the nine physiological indicators, including leaf transpiration and photosynthetic rate, stomatal conductance, root activity, and soluble protein and sugar, showed similar trends of gradually declining waterlogging stress and gradual waterlogging recovery, with little difference. However, the leaf conductivity and super oxide dismutase (SOD) activity gradually increased during flooding recovery and decreased in recovery. The tricarboxylic acid (TCA) cycle is essential for plants to grow and survive and plays a central role in the breakdown, or catabolism, of organic fuel molecules, also playing an important biological role in waterlogging stress. In total, 591 potential candidate genes were identified, and 13 particular genes (e.g., isocitrate dehydrogenase (*IDH*), malate dehydrogenase (*MDH*), ATP citrate lyase (*ACLY*), succinate dehydrogenase (*SDH*), and fumarase (*FumA*)) in the TCA cycle were also tested using qPCR. This study identifies potential candidate genes and provides theoretical support for the breeding, genetic improvement, and enhancement of the root bark yields of *P. ostii*, supporting an in-depth understanding of the plant’s physiological and molecular response mechanisms to waterlogging stress, helping future research and practice improve plant waterlogging tolerance and promote plant growth and development.

## 1. Introduction

*Paeonia ostii* (*P. ostii*) belongs to the tree peony group of the Paeoniaceae family, serving as a traditional Chinese ornamental plant, presenting both aesthetic and significant medicinal value [1,2]. *Paeonia* root bark is highly esteemed in traditional Chinese medicine for its diverse analgesic, antipyretic, antiallergic, anti-inflammatory, and immune-boosting medicinal properties. Additionally, *Paeonia* root bark is utilized as an antidepressant. Paeoniflorin, a monoterpene glucoside, is the most important medicinal compound of root bark, only being found in the *Paeonia* plants, such as *P. ostii*. Paeonol, the primary active component in *P. ostii* plant root bark, is also pivotal in evaluating cortex moutan quality. Widely used to treat cardiovascular ailments, paeonol aids in relieving tendon discomfort, promoting blood circulation, reducing heat, and cooling the blood [3]. Furthermore, damphenol serves as an additive in various daily chemical products such as toothpaste, soap, and eau de toilette, presenting extensive market potential in the food preservation and healthcare areas. *P. ostii* boasts a well-developed root system predominantly comprising fleshy roots, but it exhibits a low tolerance to waterlogged conditions. In recent years, with abnormal global climate changes, extreme weather events such as heavy rainfall have occurred frequently. If drainage is not smooth or the underground water level is too high, flood disasters occur, causing huge economic losses. In the middle and lower reaches of the Yangtze River in China, serious waterlogging occurs easily, causing flooding and flood disasters which pose a serious threat to the popularization of oil tree peonies in these areas. Tree peonies show obvious inadaptation in these areas, with frequent insect infestations and serious root rot underground, greatly affecting their ornamental value and application scope in southern China. Studies show that oil accumulation in plants causes some triglycerides to decompose in flooded environments [4]. The moisture tolerance of oil tree peony is closely related to its yield (oil content and oil yield). Woody plants occupy 31% of the Earth’s land area, about 4 billion hectares, comprising the foundation of terrestrial ecosystems. Therefore, the study of woody plants is of great economic and ecological value [5]. However, compared with herbaceous plants, few studies exist on the waterlogging or moisture tolerance of woody plants, and the induction and formation mechanisms of adventitious roots in woody plants remain unclear. Whether the mechanisms are the same as for herbaceous plants requires further study. *P. ostii* itself has ornamental, oil, and medicinal value and is of great research value, presenting a good woody plant material for studying woody plant mechanisms under flooding stress [6].

Waterlogging presents a significant challenge to sustainable agricultural development on a global scale, being exacerbated by climate change impacts and the increased frequency of extreme precipitation events. Global crop yield losses caused by waterlogging often rival those caused by drought [7]. Waterlogging can be classified into two categories based the inundation depth: waterlogging and inundation. Waterlogging occurs when water covers the soil surface, enveloping plant roots, while inundation describes partial or complete submersion of the entire plant [8]. Waterlogging’s detrimental effects on plants stem from hypoxia, primarily originating from the soil–water environment [9]. Hypoxia alters the physiological and biochemical characteristics crucial for plant growth under waterlogging stress [10]. This oxygen deficiency impacts the soil microbial community [11] and processes, leading to changes in the soil chemistry, characterized by a reduction in oxidizing substances (e.g., NO^3−^, Fe^3+^, Mn^2+^, NH_4_^+^, and SO_4_^2−^) and an increase in reducing substances (e.g., Fe^2+^, alkanes, acids, and carbonyls) [9,12,13]. Waterlogging stress influences plants’ morphological and structural traits, such as the enlargement of lenticels in woody plants [14]. While lenticels may play a role in oxygen transport and metabolism, their precise mechanism and association with plant moisture tolerance remain unclear [15]. During plant waterlogging stress, adventitious root development on plant stems serves as a crucial adaptive response [16]. In recent years, significant advancements have been made in research on the molecular mechanisms governing waterlogging tolerance. Detailed molecular-level investigations have analyzed waterlogging and waterlogging adaptation in rice [17], while successful enhancements in waterlogging tolerance in barley were achieved by exploring hypoxia-sensing signals in *Arabidopsis thaliana* [18]. Two key molecular regulatory mechanisms for survival strategies under waterlogging stress have been identified in rice: the waterlogging tolerance gene *SUBMERGENCE 1A-1* (*SUB1A*) and the escape gene (*SNORKEL1/SNORKEL2*) [19,20]. There has also been notable progress in understanding the oxygen-sensing mechanisms related to ethylene response factors *VII Ethylene Response Factor genes* (*ERF-VIIs*) [21,22,23]. Exploring waterlogging tolerance in natural wildlife species has revealed previously unknown tolerance mechanisms and genes [24,25]. However, gaps in understanding hypoxia-sensing, signaling, and downstream response mechanisms remain, with improving root aeration, hypoxia metabolism, and recovery regulation mechanisms post-stress standing out as the primary research challenges. The ornamental value and application scope of peonies in southern China are significantly impacted by strong rains and waterlogging stress. Publication of the *P. ostii* genome map brought peony scientific research into the genomic era [26], and transcriptome sequencing technology serves as a comprehensive and efficient analytical tool for *P. ostii* research, helping to uncover the molecular mechanisms underlying the essential *P. ostii* biological processes. This technology also provides technical capabilities and theoretical foundations for *P. ostii* breeding and genetic enhancement [27].

In this study, we comprehensively analyzed root system changes, explored physiological and biochemical tree peony indices under normal, waterlogged, and waterlogged recovery conditions, and identified more candidate genes for waterlogging stress. In the early sampling stage of waterlogged stress and recovery, the sampling time points were numerous and dense, and the key stage of treatment was the early stage, where plant reaction was most sensitive and rapid. In addition, samples were taken at the early and late flooding stages to comprehensively analyze the *P. ostii* response process to flooding stress. To provide a theoretical basis for improving the yield and breeding of oil peony and reveal moisture tolerance mechanisms in woody plants, we investigated the phenotypic adaptability mechanisms of leaves and root organs and explored the physiological changes in *P. ostii* under water stress.

## 2. Results

### 2.1. The Root Number and Root Tip Cell Morphology Underwent Significant Changes over the Course of the Waterlogging and Waterlogging Recovery Treatment

The root number and root tip cell morphology underwent significant changes over the course of the waterlogging and waterlogging recovery treatment (Figure 1A–F). The root number is directly related to the root vigor. Following the waterlogging stress, the longer the waterlogging time was, the lower the number of roots kept, with only 50% of roots left after W72 h, at most 25% of roots kept after W120 h, and only a few roots (<5%) left after W168 h (Figure 1A,F). This indicates that the *P. ostii* roots obviously declined during the waterlogging stress. Concurrently, the root tip cells exhibited water absorption and swelling, with the root tip color progressively darkening and eventually turning black, indicating decay and a decline in root activity. Microscopic examination revealed an increase in cell volume post-waterlogging. Furthermore, an increase in cellular staining was noted at the 72 h mark of the waterlogging treatment (Figure 1B), which suggests a gradual buildup of intracellular starch granules with prolonged waterlogging. During the post-waterlogging recovery phase, the root number and size of root tip cells gradually returned to half of the normal level, and new roots gradually increased (Figure 1D–F), suggesting the roots’ vigor and uptake capacity were partially recovered.

### 2.2. Nine Physiological and Biochemical Indicators Under Waterlogging Treatment and Recovery Conditions 

The physical parameters of the net photosynthetic rate, transpiration rate, intercellular carbon dioxide concentration, and stomatal conductance of *P. ostii* leaves gradually declined at W168 h (*p* < 0.01), and gradually increased with WR96 h (*p* < 0.01) (Figure 2A–D), with a reverse trend in leaf conductivity (Figure 2E). Specifically, the net photosynthetic rate exhibited a significant decrease (*p* < 0.05) from 6 to 72 and to 168 h of waterlogging, reaching a 41.7% reduction after 72 h. However, within 48 h of flood recovery commencing, it exhibited a nearly 80% increase compared to WR 0 h, essentially being restored to its pre-flooded state by 96 h (*p* > 0.05) (Figure 2A). A similar trend was observed for the transpiration rate, which significantly decreased (*p* < 0.05) within 72 h of waterlogging and dropped by 73.4% after 168 h. Within 48 h of flood recovery, it was quickly restored (Figure 2B). Meanwhile, the intercellular CO_2_ concentration remained statistically unchanged for *P. ostii* after 24 h of waterlogging but exhibited substantial decreases of 26.2% and 39.3% after 120 h and 168 h of waterlogging, respectively (*p* < 0.01). The levels remained non-significant during the initial 6 h of flood recovery compared with those in the pre-recovery period, with a significant increase (*p* < 0.01) noted after 48 h of recovery (Figure 2C). Stomatal conductance exhibited a gradual decline post-waterlogging treatment, decreasing by 78.8% after 168 h. During the recovery phase, stomatal conductance increased by 33.5% and 39.3% after 48 h (*p* < 0.05) and 96 h, respectively (*p* < 0.01) (Figure 2D).

Relative leaf conductivity can reflect changes in membrane permeability. The higher relative leaf conductivity, the higher membrane permeability. Leaf conductivity showed a reverse trend during waterlogging stress, exhibiting the highest value at W168 h (Figure 2E) and returning to normal during waterlogging recovery (Figure 2E).

Root soluble protein, soluble sugar, and root activity showed gradually declining trends after 168 h of flooding and were gradually partly restored after 96 h of flood (Figure 2F,G). Both soluble protein and soluble sugar displayed similar trends, showing significant reductions of 42.8% and 40.7% (*p* < 0.01) after 168 h post-waterlogging and returning to pre-waterlogging levels within 6 h of recovery (Figure 2F,G). The change trends in root activity during either W or WR stress were more sensitive than those of the root soluble protein and soluble sugar. The root activity trends underlying waterlogging and waterlogging recovery reflect the trends in root number and root tip cell in either W or WR stress, indicating that the *P. ostii* root system is a very vulnerable and sensitive organ during waterlogging stress (Figure 2H). 

The plant’s response to waterlogging-induced damage was characterized by an accelerated protective enzyme system, and a relatively smooth of superoxide dismutase (SOD) activity curve was observed in this study; however, the SOD value showed gradual enhancement from 6 to 168 h of flooding compared to the control level (*p* < 0.05), and an increase was observed after 6 h of waterlogging recovery (*p* < 0.05, Figure 2I). SOD activity significantly increased post-waterlogging treatment, with a rapid rise followed by fluctuating trends during recovery, peaking at an increase within 6 h (*p* < 0.05) (Figure 2I). 

### 2.3. A Time Sequence Analysis Unveiled Hundreds of Candidate Genes Related Waterlogging Stress Across Different Temporal Stages

Using a time sequence clustering analysis with selected DEGs, a total of twelve clusters with similar gene expression patterns were identified (Figure 3A). Notably, the Cluster 12 genes displayed gradually decreased expression during waterlogging treatment and gradually rebounded during the waterlogging recovery phase until they returned to pre-treatment levels. Their gene expression pattern trend was similar to seven of the nine mentioned physical and biochemical indices, with four leaf indices including “photosynthetic rate”, “transpiration rate”, “intercellular CO_2_ concentration”, and “stomatal conductance”, and the three root indices “soluble protein”, “soluble sugar”, and “root activity” (Figure 2). Cluster 12 comprises scanning genes involved in the negative relation to “leaf conductivity” or “root SOD activity”. Consequently, a total of 10,399 genes in Cluster 12 were selected for further targeting of genes involved in the waterlogging stress response.

Further Kyoto Encyclopedia of Genes and Genomes (KEGG) enrichment analysis of these 10,399 genes narrowed down the potential genes related to waterlogging stress, and the top 20 of 65 (Appendix A) enrichment pathways were selected, including “primarily in metabolism”, “unclassified signaling and cellular processes”, and “carbohydrate metabolism pathways” (Figure 3B). 

Otherwise, 10 of the 65 pathways with hundreds of candidate genes that were most related to waterlogging stress according KEGG annotation were explored (Figure 3C, Appendix A). Firstly, “the ubiquinone and other terpenoid quinone biosynthesis pathway” is crucial for hydrogen transfer by ubiquinone in the respiratory chain. Most importantly, “the tricarboxylic acid (TCA) cycle” serves as a pivotal component in the respiratory process. The TCA cycle is essential for plant growth and survival, playing a central role in the breakdown, or catabolism, of organic fuel molecules, i.e., glucose and some other sugars, fatty acids, and some amino acids. It also provides energy for cells. Interestingly, in total, 84 genes were involved in TCA cycle pathways, such as isocitrate dehydrogenase (*IDH*), citrate synthase (*CS*), succinate dehydrogenase (*SDH*), ATP citrate lyase (*ACLY*), malate dehydrogenase (*MDH*), and dihydrolipoamide S-succinyltransferase (*DLST*), which exhibited a gradual decrease in expression during waterlogging and a subsequent gradual increase during recovery, indicating a high involvement in the stress response (Figure 3D, Appendix A). Given the strong correlation between plant respiration, transpiration, and intracellular CO_2_ concentration, the tricarboxylic acid (TCA) cycle emerges as a critical pathway. Furthermore, various pathways involved more than 286 genes with crucial roles in plant photosynthesis, such as the “glycan biosynthesis and metabolism pathway”, “phenylalanine, tyrosine, and tryptophan biosynthesis pathway”, “porphyrin and chlorophyll metabolism pathway”, “betalain biosynthesis pathway”, and “carotenoid biosynthesis pathway”. About 185 genes in “starch and sucrose metabolism” were highly related to the plant soluble sugar content, while 110 genes in “cysteine and methionine metabolism pathway” influenced plant super oxide dismutase (SOD) activity. Acclimatized to the low-oxygen condition, plants activate gene-encoding enzymes for anaerobic pathways such as Pyruvate decarboxylase (*PDC*), Alcohol dehydrog-enase (*ADH*), aldehyde dehydrogenase (*ALDH*), plant cysteine oxidase (*PCO*), etc. [28]. Notably, the glycosphingolipid biosynthesis pathways play a significant role in biofilm structure. In water flooding, the biofilm ensures transport, energy conversion, and information transmission continue. 

### 2.4. Weighted Gene Co-Expression Network Analysis (WGCNA) of the Co-Expression Network Reveals the Association of the Turquoise Module with Physiological Traits

By utilizing WGCNA, a co-expression network was established. The network’s construction highlighted the association between trait features and gene modules, indicating a total of eighteen expression modules (Figure 4A). Notably, the turquoise module exhibited a high significant gene–trait value correlation (at least *p* = 0.013) with eight physical and biochemical indices excluding “SOD activity”, such as ”root activity” (*r* = 0.78), “soluble protein content” (0.79), “stomatal conductance” (0.81), “intercellular CO_2_ concentration” (0.84), “soluble sugar content” (0.88), “transpiration rate” (0.9), “photosynthetic rate” (0.94), and “leaf conductivity” (−0.97). The turquoise module contained 1389 genes that were first selected for exploring candidate co-expression association genes with physiological and biochemical trait changes after waterlogging. Additionally, the black module containing 157 genes was also selected because it had the highest gene traits value (*r* = 0.81, *p* = 0.0078) and significant correlation with “SOD activity” (Figure 4A).

KEGG enrichment analysis of the turquoise (Appendix A) and black modules (Appendix A) was processed, respectively. The top 20 of 42 KEGG enrichment pathways in the turquoise module included “the oxidative phosphorylation pathway”, “the ribosome”, “genetic information processing”, and “protein families” involved in “signaling and cellular processes”, among others (Figure 4B). Additionally, ten KEGG pathways highly related to water stress according to the KEGG annotations were selected (Figure 4C). The 21 waterlogging candidate genes of the TCA cycle were identified, such as *MDH*, *FumA*, *Ferredoxin-3*, and *CFU* (Appendix A, Figure 4D). Many genes in pathways such as “oxidative phosphorylation, peroxisome, photosynthesis, cysteine and methionine metabolism, and carbon fixation in photosynthetic organisms” were identified as associated with plant photosynthesis, respiration, and stomatal conductance (Appendix A, Figure 4D). For the black module, at least five glutathione S-transferase genes identified in Glutathione metabolism (00480 in KEGG) are potentially responsible for SOD activity under waterlogging stress (Figure 4A, Appendix A).

### 2.5. Screening for Candidate Genes Under Waterlogging Stress

Many candidate functional genes were found by time series analysis (Cluster 12 with 10,399) and co-expression analysis (1389 in turquoise module and 157 in the black module), respectively. An intersectional analysis between the Cluster 12 and turquoise module yielded 591 common intersecting genes (Appendix A). Subsequently, a network diagram depicting these 591 genes was constructed (Figure 5A). In the 32 pathways revealed by KEGG enrichment analysis (Appendix A), notable enrichment was found in pathways such as “ribosome”, containing 61 genes; “oxidative phosphorylation”, including 32 genes; and the “TCA cycle”, including 13 genes, such as *MDH*, *ACLY*, *acnA*, *IDH*, *SUCA*, *DLD*, *aceE*, *DLAT*, *SDH*, and *LSC1* (Figure 5B,D, Appendix A). All of these gene’s expression trends gradually decreased after 6 to 168 h of waterlogging and were gradually restored by waterlogging recovery. 

A Gene Ontology (GO) enrichment analysis was also conducted on the 591 genes, revealing their involvement in various biological processes (BPs, 329 genes), cellular components (CCs, 58 genes), and molecular functions (MFs, 87 genes) (Appendix A). In BPs, genes were significantly enriched in “biosynthetic process” (GO:0009058, *p* = 0.003, 57 genes), “protein metabolic process” (GO:0019538, *p* = 0.000016, 37 genes), “response to biotic stimulus” (GO:0009607, *p* = 0.0065, 5 genes, such as glucose-6-phosphate isomerase (GPI), Jasmonoyl-isoleucine (JA-Ile) related to waterlogging stress), and also “response to stress” (GO:0006950), including 25 genes. In CC terms, most genes were enriched in “membrane” (GO:0016020, 41 genes) and “intracellular anatomical structure” (GO:0005575, 8 genes) (Appendix A). In MFs, genes were significantly enriched in “hydrolase activity” (GO:0016787, *p* = 0.019, including 16 genes related to water stress response; “catalytic activity” (GO:0003824, 47 genes); “binding” (GO:0005488, 37 genes), and “DNA-binding transcription factor (TF) activity” (GO:0003700), with only one TF gene, a basic leucine zipper and a W2 domain-containing protein (bZIP2) annotated (Appendix A).

To validate the accuracy of *P. ostii* transcriptome data and also consider the important biological significance of the TCA circle, 13 candidate point genes involved in the TCA cycle pathway were chosen for quantitative real-time polymerase chain reaction (qRT-PCR) validation. According to the qRT-PCR experiments, the expression patterns of 6 of these genes (MDH, SUCA, LSC1, IDH_1, DLD_2, and aceE) show closely similar expression patterns to those observed in the transcriptome data, confirming the reliability of the transcriptome analysis (Appendix A), although the expression of four of the other seven genes showed (ACLY_1, ACLY_2, DLD_1, and IDH_2) showed tiny differences, and three (DLAT, SDH, and acnA) showed minor differences at a few time points (Appendix A).

## 3. Discussion

### 3.1. P. ostii Root Growth Is Inhibited Under Waterlogging Stress

*P. ostii* root growth is notably inhibited under waterlogging stress, a common occurrence due to the root system’s high sensitivity to soil moisture content [29]. Under flooded conditions, terrestrial plant roots undergo adaptations to cope with anaerobic respiration and increased energy consumption, significantly impacting root development and ultimately restricting overall plant growth. Previous studies on tomatoes have demonstrated reduced root biomass and length in response to waterlogging stress [10]. Similarly, in the current study, *P. ostii* root growth was hindered during waterlogging treatment, with gradual restoration observed during the subsequent recovery phase. This observation is supported by several key findings. During the flooding treatment, especially after 24 h of waterlogging, the number of *P. ostii* roots significantly reduced by about half. Following flooding from 24 to 168 h, the root number dramatically declined, with 5% remaining at the end. The root number recovery power was still very weak during the flooding recovery treatment. *P. ostii* root tip cells absorbed water and swelled under waterlogging stress, and the root absorption capacity was reduced. However, the plant still showed great resistance, the number of roots was gradually recovered, and the volume of slice cells and vigor of roots increased. This was accompanied by a gradual accumulation of starch granules, whereas the root tip cell morphology partially exhibited gradual recovery during the waterlogging recovery treatment (Figure 1A). Furthermore, the *P. ostii* root tips displayed deepening color progression, ultimately turning black and decaying under waterlogging stress. Although new roots emerged from existing ones, the overall root count was low. During the waterlogging recovery treatment, the color of old roots transitioned from gray–black to a partially restored creamy white hue, alleviating root morphology to some extent (Figure 1B). Additionally, the *P. ostii* root vigor notably declined after waterlogging stress but exhibited gradual improvement during the recovery phase. These morphological and observational root characteristics were consistent with the plant’s performance under waterlogging stress (Figure 1 and Figure 2G).

Plants typically develop aeration tissues within the root system as an adaptive strategy in response to waterlogging stress. Aeration tissues can be categorized into cleavage aerated tissues, which create gas spaces through cell separation and expansion without cell death, and lysed aerated tissues, which are formed through cell death followed by lysis. Under waterlogging conditions, plants trigger the formation of lysed aerated tissues, enhancing the oxygen supply to the root system. The current research on aeration tissues predominantly focuses on staple crops such as maize and wheat [30,31].

### 3.2. P. ostii Physiological and Biochemical Indices Were Significantly Affected Under Waterlogging Stress

The physiological biochemical indices of *P. ostii* were significantly influenced under waterlogging stress, which can induce soil hypoxia or even anoxia, impacting normal tree peony plant growth. Waterlogging limits nutrient and water uptake and hinders photosynthesis. The decrease in photosynthesis was attributed to stomatal conductance and non-stomatal-related limitations. Osmoregulatory substances in plants play a crucial role in improving water relations, enhancing osmoregulation, and bolstering plant resilience in adverse environments. These substances encompass proline (Pro), soluble sugars, soluble proteins, and inorganic compounds. During waterlogging stress, the levels of osmoregulatory substances such as soluble sugars and soluble proteins increased in *P. ostii* leaves, aiding cell membrane protection and minimizing cellular damage.

Plants deploy an antioxidant defense mechanism to combat waterlogging stress effects by mitigating reactive oxygen species through various antioxidants. Resilient plants exhibit heightened antioxidant enzyme activities, swiftly responding to stressors. Reactive oxygen radicals can damage cellular components such as proteins, lipids, pigments, and DNA, affecting plant metabolic processes such as photosynthesis and photosynthetic system efficiency [32]. Although plants typically produce reactive oxygen species at low levels during normal growth, exposure to waterlogging stress elevates their concentrations, inhibiting antioxidant enzyme activity and causing oxidative damage to cellular components.

In this study, waterlogging stress inflicted cellular damage to *P. ostii*, reducing the photosynthetic rate, leaf transpiration rate, and intercellular CO_2_ concentration. While these indices gradually recovered during the post-waterlogging recovery phase, they did not fully return to their original levels. The SOD enzyme, critical for cellular defense against reactive oxygen species, exhibited a high ROS scavenging capacity under waterlogging stress, mitigating plant cell membrane damage.

Previous research has highlighted significant alterations in endogenous enzymatic and non-enzymatic antioxidant levels under waterlogging conditions. For instance, wood bean genotypes displayed increased superoxide dismutase (SOD), ascorbate peroxidase (APX), catalase (CAT), and peroxidase (POD) activities under waterlogging stress [33], while mung bean plants exhibited decreased enzymatic antioxidants activities, such as catalase (CAT), ascorbate peroxidase (APX), superoxide dismutase (SOD), and glutathione reductase (GR) [34]. These studies underscore the varied responses of different plants in employing antioxidant defense systems to counteract reactive oxygen species damage under waterlogging conditions. In this study, the SOD activity trend in *P. ostii* under waterlogging stress was relatively constant, with little significant change during flooding from 6 to 72 h and no continually significant change from 72 to 168 flooding hours, suggesting that root vigor was severely damaged within 72 h of flooding stress. The *P. ostii* leaf conductivity increased during waterlogging stress and decreased post-recovery, but it did not fully return to its original level, indicating heightened cytoplasmic membrane permeability under waterlogging stress.

### 3.3. The P. ostii Tricarboxylic Acid (TCA) Cycle Is Inhibited Under Waterlogging Stress

RNA-seq technology is valuable for elucidating changes in plant transcriptomes under waterlogging conditions, providing insights into plant tolerance mechanisms [35]. During waterlogging stress, all pathways are either directly or indirectly intricately linked to the TCA cycle [36]. Notably, genes such as *IDH* and *MDH*, pivotal in the tricarboxylic acid (TCA) cycle, are significantly downregulated under waterlogging stress, with their expression gradually rising during waterlogging recovery [36]. Consistently, other important genes associated with the TCA cycle pathway, including *IDH*, *MDH*, *ACLY*, *SDH*, and *FumA*, displayed reduced expression levels in the later stages of waterlogging compared with the initial waterlogging and recovery phases. However, their expression levels gradually increased during the waterlogging recovery phase, underscoring the crucial role of these genes in plant responses to waterlogging stress. 

In summary, using advanced transcriptome sequencing technology combined with physical and biochemical indices, we identified 591 waterlogging response gene resourcing. Moreover, several differentially expressed genes related to waterlogging resistance in *P. ostii* provide new molecular markers and technical support for waterlogging resistance improvement in *P. ostii* varieties. These candidate genes all provide a strong scientific basis for *P. ostii* breeding for waterlogging tolerance and help screen waterlogging-tolerant varieties. Secondly, the physiological, biochemical, and molecular responses of *P. ostii* to waterlogging stress were explored, enriching the plants’ response mechanism to waterlogging stress and broadening the research scope of waterlogging-tolerant crop improvement. Finally, this study provides new ideas for managing agricultural waterlogging caused by climate change, showing important practical application value. It can help to promote the sustainable development of *P. ostii* under complex climate conditions and also presents a reference framework for the research and improvement of other cash crops’ waterlogging resistance.

## 4. Materials and Methods

### 4.1. Plant Material

This experiment was conducted in an artificial climate chamber at the Shanghai Chen Shan Botanical Garden. Three-year-old *P. ostii* seedlings were planted in plastic pots of 25 cm in height and 23 cm in diameter which were filled with a uniform grass charcoal substrate from Germany. Each *P. ostii* seedling was transplanted into an individual pot. The artificial climate chamber settings were maintained at temperatures of 23 °C (day) and 16 °C (night), the photosynthetic photon flux density was 500 µmol·m^2^·s^−1^, and there was a 10 h light duration per day and an air humidity of 60%. Moisture was managed using the weighing method for rehydration to sustain the relative soil moisture content. Rehydration was carried out once daily at 18:00. In total, 172 experimental seedlings were divided into three groups, namely, the control, waterlogging, and waterlogging recovery groups. Each seedling was sequentially numbered. In general, three biological replicates and three technical replicates were used for both waterlogging and waterlogging recovery groups at each sampling point. While only three biological replicates were used in the control group, no technical replicates were used.

Daily water replenishment was performed by weighing the pots to maintain a relative soil water content of 60% in the control group. For the flooded group, the potted seedlings were directly submerged in water within a pre-prepared sink, and the waterlogging treatment samples were collected at designated intervals (T0 h/T1, W6 h/T2, W24 h/T3, W72 h /WR0 h/T4, W120 h/T5, and W168 h/T6). After 3 days of waterlogging treatments, 50% of seedlings were randomly selected, restored to a 60% relative water content, and maintained at this level. Then, the waterlogging recovery samples were collected at specified intervals during the recovery phase (W72 h /WR0/T4, WR6 h/T7, WR48 h/T8, and WR96 h/T9).

### 4.2. Indicator Measurement

#### 4.2.1. Scanning Morphology of the Root System

The process began by sieving the pot soil through a 1 mm-pore-size nylon mesh. The muddy water and residual soil were filtered through a fine gauze, which was then thoroughly washed with distilled water until all soil particles were removed. Following this, root images were captured and analyzed using the WinRHIZO PRO STD4800 model root image analysis system (Regent Instruments Québec, Canada). Three biological replicates were used.

#### 4.2.2. Root Tip Slices

The paraffin slice method was employed to observe the root structure. The fresh root was immersed in Formalin-Aceto-Alcohol (FAA) and fixing solution for over 24 h, then dried using an ethanol gradient, dipped in wax, embedded, sectioned, dewaxed, and stained with safranin O-fast green before being sealed with neutral gum. Safranin O-fast green staining was done and paraffin slices were made. Three replications were carried out, examined, and photographed using a Nikon ECLIPSE Ci orthogonal fluorescence microscope (Nikon Corporation, Tokyo, Japan) and its imaging system. 

#### 4.2.3. Physical Indices of Leaf Gas Exchange Parameters

The physical indices of gas exchange parameters were tested using a Li-6400 portable photosynthesis device (LI-COR Biosciences, Lincoln, Nebraska, USA at the top of the leaves on the second and third branches of the plants. The net photosynthetic rate, transpiration rate, intercellular CO_2_ concentration, and stomatal conductance were directly recorded. 

#### 4.2.4. Determination of the Relative Leaf Conductivity

The leaf conductivity (RC) was determined with 0.5 g weighed leaves in a tube, and 20 mL of deionized water was added and vortexed for 5 min. The initial electrical conductivity was determined using a Seven2Go S7-BasicSG7-FK2 (Mettler-Toledo, Switzerland) conductivity meter and recorded as EC_1_. The leaves were then allowed to stay for 24 h at 4 °C, and the conductivity was again measured as EC_2_. The content was later autoclaved for 20 min at 121 °C and allowed to cool down. The final electrical conductivity was recorded as EC_3_. The RC was calculated using the formula *RC* = (*EC*_2_ − *EC*_1_)/(*EC*_3_ − *EC*_1_) × l00%.

#### 4.2.5. Measurement of Soluble Protein and Sugar

Soluble protein was determined by the methods described in [37], with modifications. First, 1 mL of crude enzyme solution was pipetted into a test tube, and then 5 mL of Coomassie Brilliant Blue solution (G-250) (Sharebio, Shanghai, China) was added and mixed in thoroughly; the mixture was then left to stand for 2 min. Then, the zero was adjusted with Kaumas Brilliant Blue and the absorbance value at 595 nm was determined. The calculation formula was as the following equation, protein content (mg/g) = *C* × *Vt*/(*W* × vs. × 10^3^, where *C* is the soluble protein content (μg) obtained from the standard curve, *Vt* is the total volume of the extract (mL), vs. is the volume of the aspirated sample liquid (ml), 10^3^ is the factor for the conversion of μg to mg, and *W* is the fresh weight of the sample (g).

Soluble sugar was obtained by the method [37], with modifications. The total soluble sugar concentration was determined by the anthrone colorimetric method as follows [37]. First, approximately 0.25 g root sample was homogenized using a mortar and pestle with 5 mL of distilled water and centrifuged at 1000× *g* for 10 min, and then the supernatant was collected. After appropriate dilution of the supernatant, soluble sugar was colorimetric assayed at 630 nm by adding concentrated HCl, 45% formic acid, and anthrone/sulfuric acid solution. Soluble sugar (mg/g) was calculated with the equation *C* × *Vt*/(*W* × vs. × 10^3^), where *C* is the amount of sugar obtained from the standard curve (μg), *Vt* is the total volume of the extract (mL), vs. is the volume of the aspirated sample liquid (mL), 10^3^ is the factor for the conversion of μg to mg, and *W* is the sample dried weight (g).

#### 4.2.6. Determination of the Root Superoxide Dismutase (SOD) Activity

Several clean and transparent graduated 15 mL test tubes of a uniform texture were divided into assay tubes, light control tubes, and dark control tubes, and clearly labeled. A reaction system solution (0.3 mL of 130 mmol-L^−1^ methionine solution, 0.3 mL of 0.75 mmol/L^−1^ nitroblue tetrazolium (NBT) solution, 1.5 mL of 50 mmol/L pH 7.8 phosphate buffer, and 0.5 mL of distilled water) was added to each test tube, 100 μL of the enzyme solution was added to the assay tube, and the test tubes without the enzyme solution (replaced with phosphate buffer) were treated for the light reaction. The test tubes without the enzyme solution (replaced with phosphate buffer) were used as the light control, and test tubes shielded from light using double-layered black cardboard sleeves were used as the dark control. Then, 0.3 mL of 0.02 mmol/L^−1^ riboflavin solution was added to each test tube, which was quickly placed in a 4000 lx light incubator for the light reaction at 25 °C for 30 min; the reaction was terminated by turning off the light source. The dark control was used as a blank (adjusted to zero). The absorbance value of the reaction solution of each test tube at 560 nm was measured with the following equation: SOD activity (U/g) = (*A*_0_ − *As*) × *Vt*/*A*_0_ × 0.5 × *W* × *Vs*. In this equation, *A*_0_ is the absorbance value of the control tube under light, *As* is the absorbance of the sample determination tube, *Vt* is the total volume of the sample extract (mL), vs. is the volume of the crude enzyme liquid taken at the time of determination (mL), and *W* is the fresh weight of the sample (g).

#### 4.2.7. Determination of Root Activity

The root activity was analyzed using triphenyl tetrazolium chloride (TTC). TTC is a chemical that is reduced by dehydrogenases, mainly succinate dehydrogenase, when added to a tissue. The dehydrogenase activity is regarded as an index of the root activity. In brief, 0.5 g of fresh root sample was immersed in 10 mL of equally mixed 0.4% TTC and phosphate-buffer solution, and was kept in the dark at 37 °C for 3 h. Subsequently, 2 mL of 1 mol/L H_2_SO_4_ was added to stop the reaction with the root. The root was dried with filter paper and then extracted with ethyl acetate. The red extract was transferred to a volumetric flask to reach to 10 mL by adding ethyl acetate. The absorbance of the extract was recorded at 485 nm. Root activity was expressed as TTC reduction intensity. Formally, root activity was calculated with the following equation: *a* = *r*/(*w* × *t*). Here, *a* is root activity (*μ*g/(*g* × *h*)), r is TTC reduction (*μ*g), *w* is the fresh root weight (*g*), and t is time (*h*).

#### 4.2.8. Statistical Analysis

All the data analyses were conducted with the SPSS 20.0 program (USA). The differences in the *P. ostii* root system in different waterlogging and waterlogging recovery treatment periods were analyzed using analysis of variance (ANOVA) and least significant difference (LSD) multiple comparisons. The two-tailed Student’s *t* test or one-way ANOVA was used to examine the effects of waterlogging stress or recovery treatment duration on photosynthetic rate, leaf transpiration rate, intercellular CO_2_ concentration, leaf photosynthetic rate, leaf conductivity, soluble protein, soluble sugar, root activity, and super oxide dismutase (SOD) activity. All the data were tested and they satisfied the homogeneity of variance hypothesis. * *p* < 0.05, ** *p* < 0.01, and *** *p* < 0.001.

### 4.3. RNA Extraction, Sequencing and Analysis

Plant roots were collected with two biological replications from the designated time intervals (W0 h (T1), W6 h (T2), W24 h (T3), W72 h (T4), W120 h (T5), and W168 h (T6)) after flooded stress and in the flooded recovery group (WR6 h (T7), WR48 h (T8) WR96 h (T9)). The samples were all frozen in liquid nitrogen. A total of 5 μg RNA per sample was used as input material to generate sequencing libraries using the NEBNext^®^ UltrarN^TM^ RNA Library Prep Kit for Illumina^®^ (NEB, Ipswich, MA, USA), following the manufacturer’s recommendations. After cluster generation, the libraries were sequenced on an Illumina Hiseg 2500 platform and paired-end reads were generated. 

The sequencing data were filtered with SOAPnuke (v1.5.2) and clean reads were obtained and stored in FASTQ format. Bowtie2 (v2.2.5) was applied to align the clean reads to the reference coding gene set and then the gene expression levels as FPKM (fragments per kilo base pairs per million sequencing tags) were calculated using RSEM (v1.2.12). The assembled genes were annotated by alignment to the NCBI non-redundant nucleotide (Nt) database with BLASTN and to NCBI non-redundant nucleotide protein (Nr) database using BLASTX with the option of “−e × 10^−5^”. A total of 18 samples were measured using the Illumina HiSeq platform, yielding an average of 63.09 M of data per sample (Appendix A). The average matching rate of the matched gene set was 59.40% and a total of 115,378 genes were detected (Appendix A).

Then, the log2-scaled FPKM values of genes were used to calculate the correlation coefficient matrix of samples, which was applied to the pheatmap function in the pheatmap library in R to perform hierarchical clustering of samples. The gene abundances were compared to the control to identify differentially expressed genes (DEGs) using DESeq2 (v1.4.5). Genes with an FDR < 0.05 and fold change |log2(Fold change)| > 1 were determined to be significant DEGs. Gene Ontology (GO) term and Kyoto Encyclopedia of Genes and Genomes (KEGG) pathway analysis were performed using Phyper in the R package version 4.3.1. The GO terms with multiple test-corrected *p*-values ≤ 0.05 and KEGG pathways with *p*-values ≤ 0.05 were deemed significantly enriched terms and pathways, respectively.

### 4.4. Time Sequence Analysis of the Transcriptome of the P. ostii Root System

Genes with the same expression pattern are grouped into the same cluster, and some studies use this method to analyze samples’ tissue specificity. The time series analysis in Mfuzz software v2.66 was used, based on the relaxed clustering algorithm [38].

### 4.5. WGCNA of the Transcriptome of the P. ostii Root System

Differentially expressed genes (DEGs) (FPKM > 0.5) at least one compared time point between stress and CK, either waterlogging or waterlogging recovery treatment, were used to construct a network using the WGCNA package (version 1.72). Log_2_(FPKM + 1) DEG values were used to calculate the adjacency matrix. The signed gene co-expression and nine physical and biochemical indices activity network was constructed using a soft threshold power of 7 and a minimum module size of 18 [39]. Modules with cut heights < 0.3 were merged. The gene trait value significance (correlation between gene expression pattern and nine physical and biochemical index activity change) was calculated to rank them. 

### 4.6. qRT-PCR Validation of Gene Expression

Flooded and flood-recovered *P. ostii* leaves were used for qRT-PCR experiments. The primers used for quantitative expression analysis were designed according to the cDNA sequences of the 13 selected genes involved in the TCA cycle pathway using the Primer 5 website (https://sg.idtdna.com/pages/tools/primerquest, accessed on 15 March 2024) (Appendix A). The total RNA of leaves was extracted and first-strand cDNA was synthesized using M-MuLVase. A qRT-PCR assay was carried out using an AceQ qPCR SYBRGreen Master Mix kit (Novozymes, Nanjing, China), and the reaction conditions were 95 °C pre-denaturation for 5 min and 95 °C denaturation for 10 s, and 60 °C annealing for 30 s, for 40 cycles. For melting curve acquisition, the program was 95 °C for 15 s, 60 °C for 60 s, and 95 °C for 15 s. The reaction was detected using a fluorescence quantitative PCR instrument (QuantStudio™5, Thermo Fisher Scientific, Waltham, MA, USA), and the data were analyzed using QuantStudioTM design and analysis software. The relative gene expression was calculated using the 2^−ΔΔCt^ method [40], which was repeated three times for each sample. The *P. ostii* Actin gene was used as an internal reference gene [41].

## 5. Conclusions

During waterlogging, the *P. ostii* root number and root tip cells underwent significant changes, with the root tip cortical cells expanding via water uptake, suggesting that physiological adjustments occur in response to waterlogging stress to adapt to the plant’s environmental changes. The emerging root tips’ proximity to the soil surface may be an adaptive strategy used by the plant to seek oxygen and meet respiratory requirements. Meanwhile, the yellowing, wilting, and curling of leaf margins indicates that the leaves were affected by the waterlogging stress, which can lead to the blockage of gas exchange and photosynthesis in the leaves, affecting plant growth and development. Although the basal stem, plant height, and leaf area did not show significant growth changes, a significant reduction in root vigor implied root inhibition, which can affect the plant’s nutrient uptake and water use efficiency. Under waterlogging stress conditions, plants maintain cell membrane stability by increasing the content of osmoregulatory substances such as soluble sugars and soluble proteins to mitigate the cellular damage caused by water stress. The consistency of root morphology and section observations indicates that *P. ostii* plants have strong waterlogging tolerance and recovery ability, demonstrating their adaptability under adverse conditions. As the waterlogging recovery process unfolded, the expression of genes involved in the tricarboxylic acid (TCA) cycle pathway (e.g., *IDH*, *MDH*, *ACLY*, *SDH*, and *FumA*) gradually increased, perhaps related to the gradual recovery of the normal metabolic activities in the plant, thus providing a molecular basis for plant recovery in the late stage of waterlogging stress. These results support an in-depth understanding of the physiological and molecular mechanisms of the plant response to waterlogging stress, which is helpful for future research and practice in improving plant waterlogging tolerance and promoting plant growth and development.

## Figures and Tables

**Figure 1 plants-13-03324-f001:**
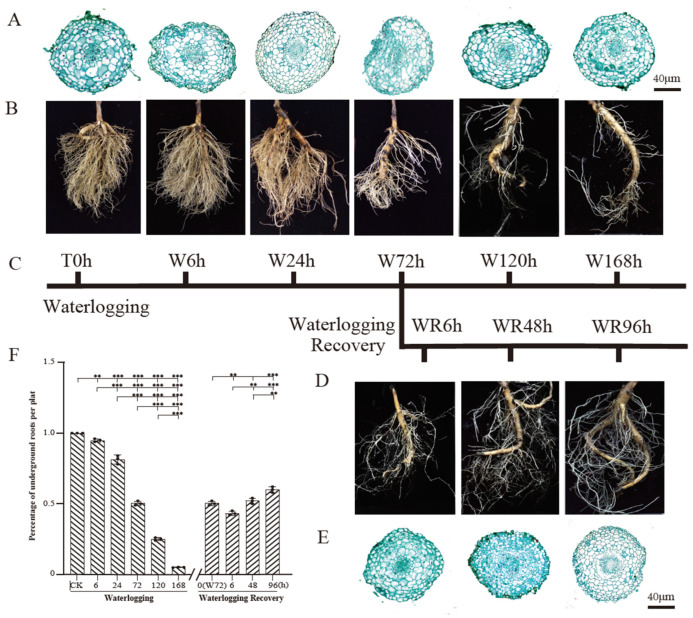
Diagram of *Paeonia ostii* root system in different periods of waterlogging and waterlogging recovery treatments. (**A**) Paraffin section of root system and (**B**) scan of root morphology with waterlogging stress at different time points. (**C**) Experimental plan and sample collection time points of waterlogging and waterlogging recovery treatments. T0 h, waterlogging start point; W, waterlogging treatment; WR, waterlogging recovery treatment; h, hour. (**D**) Paraffin section of root system and (**E**) scan of root morphology with waterlogging recovery treatment at different time points. Experimental plants for waterlogging recovery treatment were randomly selected from plants after 72 h waterlogging treatment. (**F**) Relative changes in number of roots under waterlogging and waterlogging recovery treatments. CK, waterlogging start point; 0 (W72 h), waterlogging recovery start point, equal to W72 h. Means are shown, and error bars represent the standard deviations, n = 3. Significant differences were calculated using either two-tailed Student’s *t* test or one-way ANOVA. ** *p* < 0.01, and *** *p* < 0.001.

**Figure 2 plants-13-03324-f002:**
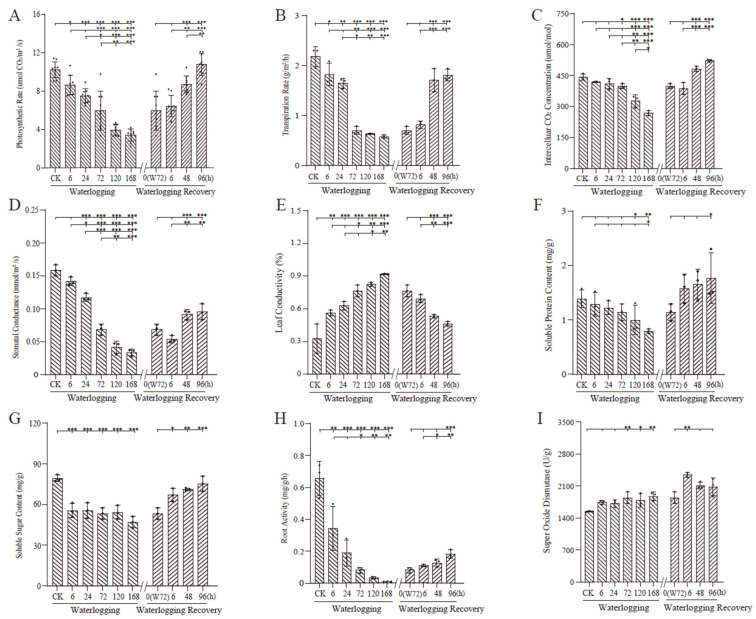
Changes in physiological and biochemical indices in waterlogging and waterlogging recovery groups for *Paeonia ostii* in different periods. Trend in photosynthetic rate (**A**), leaf transpiration rate (**B**), intercellular CO_2_ concentration (**C**), leaf photosynthetic rate (**D**), leaf conductivity (**E**), soluble protein (**F**), soluble sugar (**G**), root activity (**H**), and super oxide dismutase (SOD) activity (**I**) variation. Means are shown, and error bars represent standard deviations, n = 9 (**A**) or 3 (**B**–**I**). CK, waterlogging start point; 0 (W72 h), waterlogging recovery start point, equal to W72 h. Significant differences were calculated using either two-tailed Student’s *t* test or one-way ANOVA. * *p* < 0.05, ** *p* < 0.01 and *** *p* < 0.001.

**Figure 3 plants-13-03324-f003:**
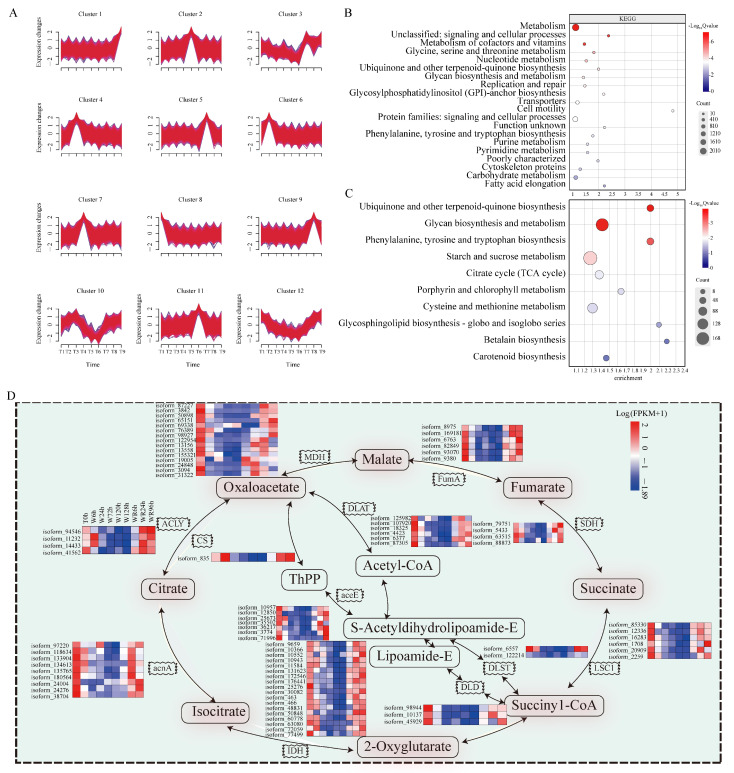
Time sequence analysis reveals important role of TCA pathway genes in waterlogging and waterlogging recovery treatments. (**A**) Time series clustering diagram. T1: T0 h; T2: W6 h; T3: W24 h; T4: W72 h (WR0 h); T5: W120 h; T6: W168 h; T7: WR6 h; T8: WR48 h; T9: WR96 h. (**B**) Top KEGG pathway enrichments of Cluster 12. (**C**) KEGG pathway enrichments related to waterlogging or waterlogging recovery stress. (**D**) Important genes of TCA cycle pathway expression pattern underlying W or WR treatments in different time points. W, waterlogging; WR, waterlogging recovery; h, hour; FPKM, fragments per kilobase of exon model per million mapped fragments.

**Figure 4 plants-13-03324-f004:**
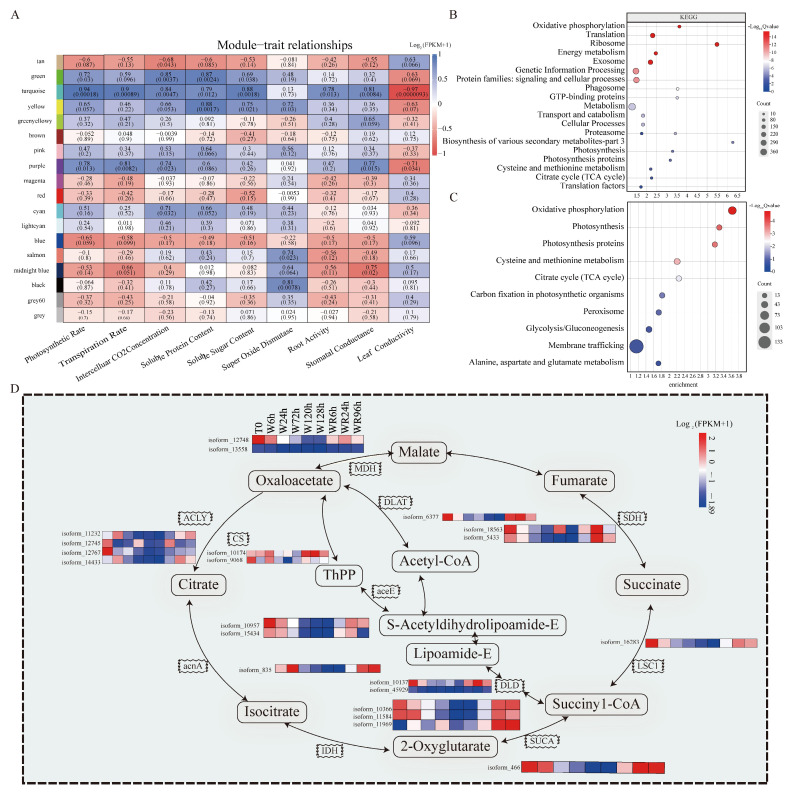
Scanning for candidate genes underlying waterlogging stress using Weighted gene co-expression network analysis (WGCNA) and KEGG analysis for RNAseq data. (**A**) Total 18 gene modules revealed by WGCNA. (**B**) Top 20 KEGG enrichment pathways in turquoise module. (**C**) Ten KEGG enrichment waterlogging stress-related pathways by annotation. (**D**) Total 21 gene involved into TCA cycle pathway in turquoise module. W, waterlogging; WR, waterlogging recovery; h, hour; FPKM, fragments per kilobase of exon model per million mapped fragments.

**Figure 5 plants-13-03324-f005:**
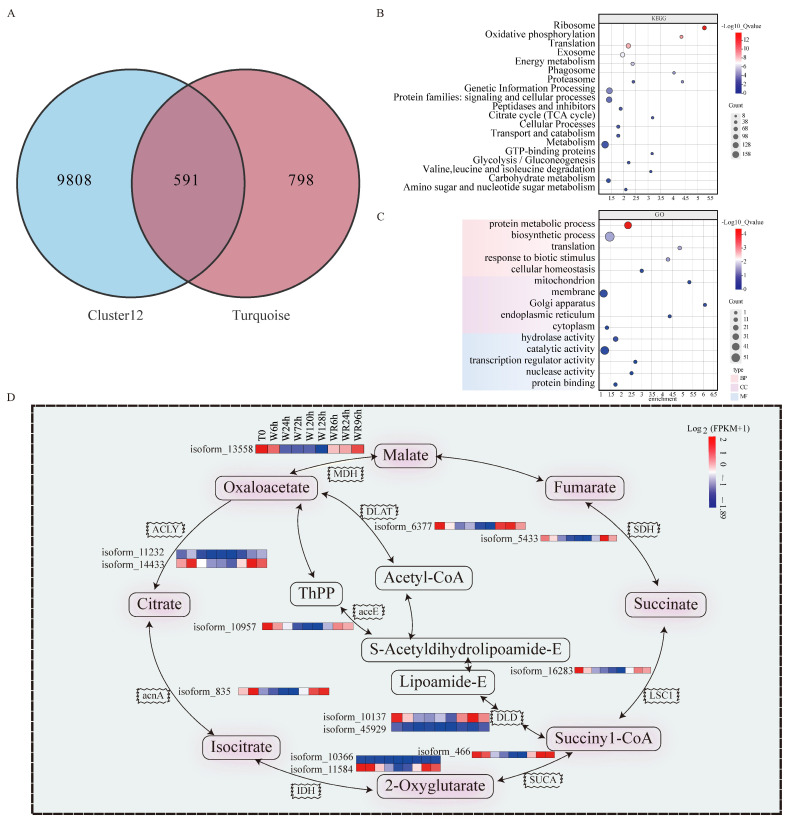
Screening for candidate genes under waterlogging stress. (**A**) Intersection genes of Cluster 12 and turquoise module. (**B**) KEGG enrichment of the common genes between Cluster 12 and turquoise module. (**C**) GO enrichment map of common genes between Cluster 12 and turquoise module. GO, Gene Ontology; BP, biological process; MF, molecular function; CC, cell component. (**D**) Pathway genes enriched in TCA cycle intersection of Cluster 12 and turquoise genes. W, waterlogging; WR, waterlogging recovery; h, hour; FPKM, fragments per kilobase of exon model per million mapped fragments.

## Data Availability

The Illumina sequencing data have been deposited in the Genome Sequence Archive under the accession number PRJNA1122327.

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
