# Peer review of "New Insight into the Related Candidate Genes and Molecular Regulatory Mechanisms of Waterlogging Tolerance in Tree Peony *Paeonia ostii"

_plants, 2024, doi:10.3390/plants13233324_

Round 1
Reviewer 1 Report
Comments and Suggestions for Authors
The study effectively summarizes the research objectives, methods, and key findings, providing a clear picture of what the study entails. The study addresses an important agricultural issue—waterlogging tolerance—which is highly relevant for improving crop yields in waterlogged regions. Some minor issues should be addressed before publication.
The abstract includes technical terms (e.g., tricarboxylic acid (TCA) cycle, specific gene names) which might be challenging for readers not familiar with plant physiology or molecular biology. Simplifying some of these terms or providing brief explanations could make the abstract more accessible.
In the results section, some sentences are quite long and complex, which can affect readability. Breaking down complex sentences into simpler, more concise statements might improve clarity.
While the abstract mentions significant changes and differences, it lacks specific quantitative data (e.g., percentage changes, p-values) that could strengthen the reported findings.
Although the study mentions the implications for future research and practice, it could benefit from a more explicit statement on what specific future research directions or practical applications could be pursued based on the study’s findings.
While it mentions changes in root tip sections and gene expression, the discussion could be more specific about how these changes relate to waterlogging tolerance and plant health overall.
Line 56-58 could cite Liu et al., 2023.Silver lining to a climate crisis in multiple prospects for alleviating crop waterlogging under future climates. Nat Commun 14, 765 (2023).
Comments on the Quality of English Languageminor revision
Author Response
R1# Comments and Suggestions for Authors
- The study effectively summarizes the research objectives, methods, and key findings, providing a clear picture of what the study entails. The study addresses an important agricultural issue—waterlogging tolerance—which is highly relevant for improving crop yields in waterlogged regions. Some minor issues should be addressed before publication.
Response:
We feel great thanks for your professional review work on our article. According to your nice suggestions, we have made extensive corrections to our previous draft and all revisions to the manuscript have been highlighted.
- The abstract includes technical terms (e.g., tricarboxylic acid (TCA) cycle, specific gene names) which might be challenging for readers not familiar with plant physiology or molecular biology. Simplifying some of these terms or providing brief explanations could make the abstract more accessible.
Response:
We sincerely appreciate the valuable comments. Now it reads,
The tricarboxylic acid (TCA) circle is essential for plants to grow and survive, which plays a central role in the breakdown, or catabolism, of organic fuel molecules, also important biological meaning in the waterlogged stress. In total of 591 potential candidate genes were identified, especially 13 genes (e.g, isocitrate dehydrogenase (IDH), malate dehydrogenase (MDH), ATP citrate lyase (ACLY), succinate dehydrogenase (SDH) and fumarase (FumA) in TCA cycle were also tested by qPCR. (Line 26-31, page 1)
- In the results section, some sentences are quite long and complex, which can affect readability. Breaking down complex sentences into simpler, more concise statements might improve clarity.
Response:
Thanks, Long and complicated sentences have been carefully checked and improved Such as the following example. Actually, the results section had been nearly rewritten again.
#1 Furthermore, an increase in cellular staining was noted at the 72-hour mark of the waterlogging treatment. It suggests a gradual buildup of intracellular starch granules with prolonged waterlogging. (Line137-139, page3)
- While the abstract mentions significant changes and differences, it lacks specific quantitative data (e.g., percentage changes, p-values) that could strengthen the reported findings.
Response:
Thank you for your kind suggestion, now it reads.
As flooding continued, roots vigor dramatically declined from 6 to 168 hours waterlogging, and the root number was extremely reduced at most 95%, while root number never being restored after recovery 96 hours. The seven of nine physiological indicators, such as leaf transpiration and photosynthetic rate, stomatal conductance, root activity, soluble protein and sugar, showed similar trends to gradually decline by waterlogging stress and gradually waterlogging recovery with little differences. However, the leaf conductivity and super oxide dismutase (SOD) activity showed trends of gradually increased during flooding recovery and decreased in recovery.
(Line19-26, page1)
- Although the study mentions the implications for future research and practice, it could benefit from a more explicit statement on what specific future research directions or practical applications could be pursued based on the study’s findings.
Response:
Thank you for your kind suggestion and this phrase was modified according to the comment. Now it reads.
In recent years, with the abnormal changes of global climate, extreme weather events such as heavy rainfall occur frequently. Once the drainage is not smooth or the underground water level is too high, it will form flood disasters, causing huge economic losses. In the middle and lower reaches of the Yangtze River in China, serious waterlogging is easy to occur, resulting in flooding and flood disaster, which poses a serious threat to the popularization of oil tree peonies in these areas. Tree peonies show obvious inadaptation in these areas, with frequent insect infestation and serious root rot underground, which greatly affects the ornamental value and application scope of tree peonies in southern China. Studies have shown that oil accumulation in plants will change direction and some triglycerides will decompose in the flooded environment[4]. The moisture tolerance of oil tree peony is closely related to its yield (oil content and oil yield). Woody plants occupy 31% of the earth's land area, about 4 billion hectares, and are the foundation of terrestrial ecosystems. Therefore, the study of woody plants is of great economic and ecological value[5]. However, compared with herbaceous plants, there are few studies on the waterlogging or moisture tolerance of woody plants, and the mechanism of induction and formation of adventitious roots in woody plants is still unclear. Whether it is the same as the case of these herbaceous plants needs to be further studied. P. ostii itself has ornamental, oil and medicinal value, and it is of great research value and provides a good woody plant material for studying the mechanism of woody plants under flooding stress[6]. (Line54-73, page2).
In this study, we comprehensively analyzed the changes of root system, physiological and biochemical indices of tree peony under normal, waterlogged and waterlogged recovery conditions and tried to identified more candidate genes for waterlogging stress. In the early sampling stage of waterlogged stress and recovery, the sampling time points were many and dense, and the key stage of treatment was the early stage where plant reaction was most sensitive and rapid. In addition, samples were taken at the early and late stages of flooding to comprehensively analyze the response process of P. ostii to flooding stress. To provide theoretical basis for improving the yield and breeding of oil peony and reveal the mechanism of moisture tolerance in woody plants, we investigated the mechanism of phenotypic adaptability such as leaves and root organs and physiological changes of P. ostii under water stress.
(Line113-123, page3).
- While it mentions changes in root tip sections and gene expression, the discussion could be more specific about how these changes relate to waterlogging tolerance and plant health overall.
Response:
Thank you for your kind suggestion and this phrase was modified according to the comment.
During the flooding treatment, especially after the 24-hour waterlogging, the number of P. ostii roots was significantly reduced by about one half. Following flooding from 24 to 168 hours, the root number dramatically declined to at most 5% left. And the root number recovery power was still very weak during flooding recovery treatment. The root tip cells of P. ostii absorbed water and swelled under waterlogging stress and the root absorption capacity was reduced, the plant still showed great resistance, the number of roots was gradually recovered, the volume of slice cells, and the vigor of roots increased. (Line337-384, page11)
Waterlogging limits nutrient and water uptake and hinders photosynthesis. The decrease in otosynthesis was attributed to stomatal conductance and non-stomatal related limitations. (Line406-408, page11)
- Line 56-58 could cite Liu et al., 2023.Silver lining to a climate crisis in multiple prospects for alleviating crop waterlogging under future climates. Nat Communication 14, 765 (2023).
Response:
Thank you for your insightful suggestion. This excellent reference has been added to manu.
- Liu K, Harrison M, Yan H, et al. Silver lining to a climate crisis in multiple prospects for alleviating crop waterlogging under future climates[J]. Nature Communication, 2023, 14(1):765.
(Line 702-703, page18).
- R1# Comments on the Quality of English Language : minor revision
Response:
Thanks, a professional language editing has been done by MDPI author service.
Reviewer 2 Report
Comments and Suggestions for Authors
The manuscript by Minghui Zhou provides insight into the response to waterlogging in Paeonia ostii.
Positives - Interesting plant species, different complementary techniques covering transcriptome, selected established markers of stress/plant fitness, and microscopy.
Negatives - The highlighted processes are well-established pathways found in waterlogged plants, and the novelty/differences (if any) are not clearly presented/compared with the established model plants. Results are poorly presented, and there are contradictions in the presentation. Materials & Methods are incomplete and poorly written.
Major issues:
-
The results of root morphology and root sections presented in the first part of the results section could be interesting but lack quantitative analyses (e.g., density, area, weight). Without that, the reproducibility is not clear, and the results do not provide sufficient biological relevance. The authors should also specify the number (N=) of biological replicates for each presented experiment.
-
The second set of results is an assortment of different parameters. Part of the presented results is not covered in the methodology, and statistical evaluation of significant differences and the number of biological/technical replicates is missing for all panels.
-
Panel C is missing units.
-
Panel E - Soluble sugar content in roots only rarely exceeds 20% DW. Here, the authors claim 8% FW to be present in the measured samples. That seems to be a significant overestimation and could indicate some issues with the methodology.
-
Panel F - Listed units (U/g/h) are unconventional and must be corrected. Furthermore, enzyme activity should be normalized to protein content, not FW.
-
Panel D - The authors highlight the differences in protein content. However, a two-fold difference in protein content (if statistically significant) only reflects a higher amount of water in waterlogged tissues, and it would be strange to find it substantially higher in WR samples compared to T0h.
-
The second part of the manuscript is predominantly based on correlation analysis using a less reliable Spearman's correlation coefficient that is prone to false positives and easily swayed by extreme values in the dataset. A more advanced non-linear mixed model would be more appropriate here as the validation of the results using an independent assay is missing.
-
The authors claim that the SOD enzyme exhibited a high ROS scavenging capacity under waterlogging stress. However, that is not shown in the manuscript. The only presented data indicate that there was no significant difference in SOD activity and that the increase in SOD was found only in recovering plants. I would like to note that the correctly normalized SOD activity to protein content could match the observation previously reported in waterlogged plants.
-
Materials and Methods are very poorly written, and there are missing sections on methods used for obtaining presented results.
- The results comparison with established plant models is missing. The presented highlights (ROS, TCA) are well-known pathways impacted by waterlogging. The authors should focus on novelty and features that were not previously described in model plants.
Minor issues:
-
The introduction contains misleading and incorrect information. H2S is not an oxidizing substance.
-
The Transcriptomics presentation lacks information on the FC threshold and statistical criteria for selecting significant differences.
-
"while cysteine and methionine metabolism influence plant superoxide dismutase (SOD) activity" — this information should be supplemented with a reference.
-
The sentence in lines 201-204 does not seem to be well integrated into the manuscript.
-
Fig. 4 - Fonts are too small, and gradients for heatmaps are not correctly described in the legend.
-
qPCR validation seems more suitable for supplementary data. It is missing statistical evaluation, and the claim that it is similar to the NGS results is not entirely correct (panels K-M).
-
Genes should always be italicized (e.g., L358 - MDH, IDH).
-
500 μmol·m2·s-1 is PPFD, not light intensity.
-
Kaumas Brilliant Blue G-250 does not exist (Methods).
-
Supplementary information lists identical titles for Tables S1 and S2. The title is misleading: "Genes involved in the tricarboxylic acid (TCA) cycle pathway according to weighted gene co-expression network analysis" — TCA cycle genes are known and do not require weighted gene co-expression network analysis.
parts of the manuscript need significant improvement
Author Response
R2# Comments and Suggestions for Authors
The manuscript by Minghui Zhou provides insight into the response to waterlogging in Paeonia ostii.
Positives- Interesting plant species, different complementary techniques covering transcriptome, selected established markers of stress/plant fitness, and microscopy.
Response:
We appreciate your summary of the manuscript and encouraging comments.
Negatives- The highlighted processes are well-established pathways found in waterlogged plants, and the novelty/differences (if any) are not clearly presented/compared with the established model plants. Results are poorly presented, and there are contradictions in the presentation. Materials & Methods are incomplete and poorly written.
Response:
We would like to thank you for your professional review work, constructive comments, and valuable suggestions on our manuscript. Your time and efforts are greatly appreciated. We have strived to improve all the issues as you mentioned in manu, all Figures and additional Tables. All of the manuscript improvements have been highlighted as green color.
Major issues:
- The results of root morphology and root sections presented in the first part of the results section could be interesting but lack quantitative analyses (e.g., density, area, weight). Without that, the reproducibility is not clear, and the results do not provide sufficient biological relevance. The authors should also specify the number (N=) of biological replicates for each presented experiment.
Response:
We would like to thank the reviewer for pointing out this issue and have updated the text as suggested.
Now it reads.
The root number and root tip cell morphology underwent significant changes over the course of the waterlogging and waterlogging recovery treatment (Figure 1A, 1B, 1C, 1D, and 1E). The root number is directly related to the root vigor. Following the waterlogging stress, the longer waterlogging time, the less root number kept, only 50% root left after W72 h, and at most 25% root kept after W120 h, and only few rootS (< 5%) left after W168 h (Figure 1A). It indicated that the P. ostii roots were obviously declining during waterlogging stress. Concurrently, the root tip cells exhibited water absorption and swelling, with the color of the root tip progressively darkening and eventually turning black, indicating decay and decline of the root activity. A microscopic examination revealed an increase in cell volume post-waterlogging. Furthermore, an increase in cellular staining was noted at the 72-hour mark of the waterlogging treatment (Figure 1B). It suggests a gradual buildup of intracellular starch granules with prolonged waterlogging. During the recovery phase post-waterlogging, the root number and size of root tip cells gradually returned to half of normal level, and new roots gradually increased (Figure 1D and 1E), suggesting the roots vigors and uptaking capacity partially were recovered.(In Results Section, Line127-141, page 3).
While only three biological replicates were used in the control group, no technical replicates were used. (In Methods Section, Line 486-487, page13)
Following this, root images were captured and analyzed using the WinRHIZO PRO STD4800 model root image analysis system from Regent, Canada. Three biological replicates were used. (In Methods Section, Line 500-502, page13).
- The second set of results is an assortment of different parameters. Part of the presented results is not covered in the methodology, and statistical evaluation of significant differences and the number of biological/technical replicates is missing for all panels.
Response:
We thank the reviewer for bringing this to our attention. As you are concerned, there are several problems that need to be addressed. According to your nice suggestions, we have made extensive corrections to our previous draft, the Methods Section has been thoughtfully revised and supplemented, including biological replicate and statistical significance methods, and the detailed corrections are listed below.
4.1. Plant Material
The experiment was conducted in the artificial climate chamber at the Shanghai Chen Shan Botanical Garden. The 3-year-old P. ostii seedlings were planted in the plastic pots with 25 cm in height and 23 cm in diameter which filled with a uniform grass charcoal substrate from Germany. Each P. ostii seedling is transplanted into an individual pot. The artificial climate chamber settings were maintained at temperatures of 23°C (day) and 16°C (night), the photosynthetic photon flux density was 500 µmol·m2·s-1, a 10-hour light duration per day, and an air humidity of 60%. Moisture was managed using the weighing method for rehydration to sustain the relative soil moisture content. Rehydration was carried out once daily at 18:00. In total, 172 experimental seedlings were divided into three groups, namely, control, waterlogging, and waterlogging recovery groups. Each seedling sequentially was numbered. In general, three biological replicates and three technical replicates were used for both waterlogging and waterlogging recovery group at each sampling point. While only three biological replicates were used in the control group, no technical replicates were used.
Daily water replenishment was performed by weighing the pots to maintain a relative soil water content of 60% in the control group. For the flooded group, the potted seedlings were directly submerged in water within a pre-prepared sink, and the waterlogging treatment samples were collected at designated intervals (T0h/T1, W6h /T2, W24h/T3, W72h/ WR0h/T4, W120h/T5, and W168h/T6). Random selecting 50% seedlings after 3 days of waterlogging treatments, and restored to a 60% relative water content, and maintained at this level. Then the waterlogging recovery samples were collected at specified intervals during the recovery phase (WR6h/T7, WR48h/T8 and WR96h/T9). (Line473-495, page13)
Following this, root images were captured and analyzed using the WinRHIZO PRO STD4800 model root image analysis system from Regent, Canada. Three biological replicates were used.
4.2.2. Root tip slices
The paraffin slice method was employed to observe the root structure. The Fresh root was then immersed in the Formalin-Aceto-Alcohol (FAA), and fixing solution for over 24 h, dried using an ethanol gradient, dipped in wax, embedded, sectioned, and, following dewaxing, stained with safranin O-fast green before being sealed with neutral gum. Safranin O-fast green staining and paraffin slices were made. Three replications were carried out and it was examined and photographed using a Nikon ECLIPSE Ci orthogonal fluorescence microscope (Nikon Corporation, Tokyo, Japan) and its imaging system. (Line500-510, page13)
4.2.4. Determination of the relative leaf conductivity
The leaf conductivity (RC) was determined with 0.5 g weighed leaves in a tube, and 20 mL of deionized water was added and vortexed for 5 min. The initial electrical conductivity was determined using Seven2Go S7-BasicSG7-FK2 (Mettler-Toledo, Switzerland) conductivity meter, and recorded as EC1. These were then allowed to stay for 24 h at 4 ◦C, and the conductivity was again measured as EC2. The content was later autoclaved for 20 min at 121 ◦C and allowed to cool down. The final electrical conductivity was recorded as EC3. The RC was caculated using the formula, RC = (EC2 - EC1) / (EC3 - EC1) x l00%. (Line516-523, page13-14)
4.2.5. Determination of the root soluble protein content
The soluble protein was determined using Kaumas Brilliant Blue G-250 (Sharebio, Shanghai, China) as follows. First, 1 ml of crude enzyme solution was pipetted into a test tube, and then 5 ml of Kaumas Brilliant Blue G-250 reagent was added and mixed in thoroughly; the mixture was then left to stand for 2 min. Then, the zero was adjusted with Kaumas Brilliant Blue, and the absorbance value at 595 nm was determined. The calculation formula was as the following equation, Protein content (mg/g) = C x Vt / (W x Vs x 103) x 100%, where C is the soluble protein content (μg) obtained from the standard curve, Vt is the total volume of the extract (ml), Vs is the volume of the aspirated sample liquid (ml); and W is the fresh weight of sample (g).
4.2.6. Determination of the root soluble sugar content
The total soluble sugar concentration was determined by anthrone colorimetric method as follows. First, approximately 0.25 g root sample was homogenized using a mortar and pestle with 5 ml of distilled water, then centrifuged at 1,000 ×g for 10 min, the supernatant was collected. After appropriate dilution of the supernatant, soluble sugar was colorimetric assayed at 630 nm by adding concentrated HCl, 45% formic acid, and anthrone/sulphuric acid solution. Percent soluble sugar (mg/g) was calculated as the following equation, C x Vt (W x Vs x 106) x 100%, where C is the amount of sugar obtained from the standard curve (μg), Vt is the total volume of the extract (ml), Vs is the volume of the aspirated sample liquid (ml), 106 is the factor for the conversion of g to μg, and W is the sample weight (g).
4.2.7. Determination of the root superoxide dismutase (SOD) activity (Line524-546, page14)
The calculation is following the equation, SOD activity(U/g)= (A0 - As) × Vt / A0 × 0.5 × W × Vs, where A0 is the absorbance value of the control tube under light, As is the absorbance of the sample determination tube, Vt is the total volume of the sample extract (ml), Vs is the volume of the crude enzyme liquid taken at the time of determination (ml), and W is the fresh weight of the sample (g).
4.2.8 Determination of root activity
Root activity was analyzed by the triphenyl tetrazolium chloride (TTC). TTC is a chemical that is reduced by dehydrogenases, mainly succinate dehydrogenase, when added to a tissue. The dehydrogenase activity is regarded as an index of the root activity. In brief, 0.5 g fresh root samples was immersed in 10 ml equally mixed solution of 0.4% TTC and phosphate buffer, and kept in the dark at 37 ℃ for 3 h. Subsequently, 2 mL of 1 mol/L H2SO4 was added to stop the reaction with the root. The root was dried with filter paper and then extracted with ethyl acetate. The red extract was transferred to volumetric flask to reach to 10 mL by adding ethyl acetate. The absorbance of the extract at 485 nm was recorded. Root activity was expressed as TTC reduction intensity. Formally, root activity was calculated with the following equation, a = r / (w × t), where a is root activity (μg / (g × h)), r is TTC reduction (μg), w is the fresh root weight (g), and t is time (h).
4.2.9. Statistical Analysis
All the data analyses were conducted with SPSS 20.0 program (USA), either two-tailed Student’s t test or two-way ANOVA compare the statistic significant difference between the data with CK (control group) or between waterlogging and waterlogging recovery treatments. *P < 0.05, **P < 0.01.
4.3 RNA extraction, sequencing and analysis
Plant roots were collected with two biological replications from the designated time intervals (W0h (T1), W6h (T2), W24h (T3), W72h (T4), W120h (T5), and W168h(T6)) after flooded stress and in the flooded recovery group (WR6h (T7), WR48h (T8) WR96h (T9)). Those samples all frozen in liquid nitrogen. A total of 5 μg RNA per sample was used as input material to generate sequencing libraries using the NEBNext® UltrarNTM RNA Library Prep Kit for llumina® (NEB, USA) following the manufacturer's recommendations. After cluster generation, the libraries were sequenced on an Illumina Hiseg 2500 platform, and paired-end reads were generated.
The sequencing data was filtered with SOAPnuke (v1.5.2), and clean reads were obtained and stored in FASTQ format. The clean reads were mapped to reference full-length transcriptome using HISAT2 (v2.0.4). Bowtie2(v2.2.5) was applied to align the clean reads to the reference coding gene set and then the gene expression levels as FPKM (Fragments Per Kilo basepairs per Million sequencing tags) were calculated using RSEM (v1.2.12). The assembled genes were annotated by being aligned to the NCBI non-redundant nucleotide (Nt) database with BLASTN and NCBI non-redundant nucleotide protein (Nr) database using BLASTX with the option of “-e 1E-5”. A total of 18 samples were measured using an Illumina HiSeq platform, yielding an average of 63.09 M of data per sample (Table S5). The average matching rate of the matched gene set was 59.40%; a total of 115,378 genes were detected (Table S11).
Then the log2-scaled FPKM values of genes were used to calculate the correlation coefficient matrix of samples which was applied to the pheatmap function in the pheatmap library in R to perform hierarchical clustering of samples. The abundances of genes were compared to control to identify Differentially Expressed Genes (DEGs) using the DESeq2 (v1.4.5). Genes with a FDR < 0.05 values and fold change |log2(Fold change)| >1 values were used as the significantly DEGs. Gene Ontology (GO) term and Kyoto Encyclopedia of Genes and Genomes (KEGG) pathway using Phyper in the R package version 4.3.1. The GO terms with multiple test corrected P-values ≤0.05 and KEGG pathways with P-values ≤0.05 were regarded as significantly enriched terms and pathways, respectively.
4.4. Time sequence analysis of the transcriptome of the P. ostii root system
Genes with the same expression pattern are grouped into the same cluster, and some studies use this method to analyze the tissue specificity of the samples. The time series analysis Mfuzz software were used, which is based on the relaxed clustering algorithm [39].
4.5. WGCNA of the transcriptome of the P. ostii root system
Differentially expressed genes (DEGs) (FPKM > 0.5) in at least one compared time point between stress and CK, either waterlogging or waterlogging recovery treatment, were used to construct a network by using WGCNA package (version 1.72). Log2(FPKM+1) values of DEGs were used to calculate the adjacency matrix. The signed gene co-expression and nine physical and biochemical index activity network was constructed by using soft threshold power of 7 and a minimum module size of 18[40]. Modules with cut heights < 0.3 were merged. Gene trait value significance (correlation between gene expression pattern and nine physical and biochemical index activity change) were calculated to rank them. (Line559-625, page14-16)
Figure 2. Changes in physiological and biochemical indices in waterlogging and waterlogging recovery groups for P. ostii in different periods. Trend of variation in photosynthetic rate(A), leaf transpiration rate (B), intercellular CO2 concentration (C), leaf photosynthetic rate (D), leaf conductivity (E), soluble protein (F), soluble sugar (G), root activity (H), and super oxide dismutase (SOD) activity (I). The means are shown, and the error bars represent the standard deviations. Significant differences were calculated using either two-tailed Student’s t test or two-way ANOVA. *P < 0.05, **P < 0.01. (Line197-206, page5)
We have changed the order of Panels (in Figure 2) for more easily understood, statistic test also added into the figures.
- Panel C is missing units.
Response:
We were sorry for our careless mistakes. Thank you for your reminder. The unit has been added to the The Panel C figure as follows:
- Panel E - Soluble sugar content in roots only rarely exceeds 20% DW. Here, the authors claim 8% FW to be present in the measured samples. That seems to be a significant overestimation and could indicate some issues with the methodology.
Response:
Thank you for your kind reminders. Panel E is revised by adding the results of statistical tests. After carefully checking and reanalyzing the source data again, the results of soluble sugar content were correct (showed in Panel E before, now in Figure 2G).
The sugar contents of 8% is higher but a normal and reasonable value for tree peonies. according to several published references, such as 1) Cui H, Yu J, Gao X, et al. Study on cold resistance and physiological characteristics of three purple peony varieties [J]. Journal of Northeast Agricultural University, 2009, 40(07):24-27. (in Chinese). 2) Li X, Li T, Zhang Z, et al. Analysis of soluble sugar, soluble protein content and antioxidant enzyme activity in pollen of different Peony cultivars[J]. Zhejiang Forestry Science and Technology, 2023, 43(04):57-62. (in Chinese). 3) Li J, Guo L, Kong X, et al. Effects of 6-BA and GA3 on physiological characteristics of peonies during leaf senescence [J]. Chinese Journal of Plant Physiology, 2014, 50(08):1243-1247. (in Chinese).
(1) Cui H, Yu J, Gao X, et al. Study on cold resistance and physiological characteristics of three purple peony varieties [J]. Journal of Northeast Agricultural University, 2009, 40(07):24-27. (in Chinese).
(2) Li X, Li T, Zhang Z, et al. Analysis of soluble sugar, soluble protein content and antioxidant enzyme activity in pollen of different Peony cultivars[J]. Zhejiang Forestry Science and Technology, 2023, 43(04):57-62. (in Chinese).
(3) Li J, Guo L, Kong X, et al. Effects of 6-BA and GA3 on physiological characteristics of peonies during leaf senescence [J]. Chinese Journal of Plant Physiology, 2014, 50(08):1243-1247. (in Chinese)
(In this study)
- Panel F - Listed units (U/g/h) are unconventional and must be corrected. Furthermore, enzyme activity should be normalized to protein content, not FW.
Response:
Thank you for your remainder. The unit of Panel F has been corrected as showing:
- Panel D - The authors highlight the differences in protein content. However, a two-fold difference in protein content (if statistically significant) only reflects a higher amount of water in waterlogged tissues, and it would be strange to find it substantially higher in WR samples compared to T0h.
Response:
Thank you for your insightful comment and kind suggestion.
A mistake of y scale in Panel D figure has been corrected. The new Panel D picture (Figure 2I now) has been modified as follows:
- The second part of the manuscript is predominantly based on correlation analysis using a less reliable Spearman's correlation coefficient that is prone to false positives and easily swayed by extreme values in the dataset. A more advanced non-linear mixed model would be more appropriate here as the validation of the results using an independent assay is missing.
Response:
We thank the Reviewer for the pitifulness of the less reliable using WGCNA methods mostly dependent on the correlation coefficient. We have used intersection methods to get common genes between a time sequence clustering analysis and the WGCNA analysis to reduce the false opportunity, and we also using qPCR experiment to test the candidate genes. In my opinion, WGCNA methods still is a powerful tool to handle thousands of expression data, but we really keep a cool head, the results from WGNCA are just a correlation measures, more evidence is need for biological explanations.
- The authors claim that the SOD enzyme exhibited a high ROS scavenging capacity under waterlogging stress. However, that is not shown in the manuscript. The only presented data indicate that there was no significant difference in SOD activity and that the increase in SOD was found only in recovering plants. I would like to note that the correctly normalized SOD activity to protein content could match the observation previously reported in waterlogged plants.
Response:
Thanks for kindly reminding us to clarify this point. And the correctly normalized SOD activity to protein content has also been done.
The plant's response to waterlogging-induced damage was characterized by an accelerated protective enzyme system, a relatively smooth curve of superoxide dismutase (SOD) activity was observed in this study but the SOD value still a gradual enhancement from 6 to 168 hours flooding compared to control level (P < 0.05), and increasing observed after 6 waterlogging recovery (P < 0.05, Figure 2I). SOD activity significantly increased post-waterlogging treatment, with an rise rapid, and followed by fluctuating trends during recovery, peaking at a increase within 6 hours (P < 0.05) (Figure 2I). (Line190-197, page5).
In this study, the activity trend of SOD in P. ostii under waterlogging stress were relatively easy during flooding stress with little significant change during flooding from 6 to 72 hour, but no continually significant change from 72 to 168 flooding hours, also suggesting the root vigor dramatically damaged within 72 flooding stress. (Line437-441, page12)
- Materials and Methods are very poorly written, and there are missing sections on methods used for obtaining presented results.
Response:
Thank you for kindly reminding us. The Materials and Methods parts have been revised carefully and thoroughly, please check the previous response to the Comments 2, as well as checking in the Materials and Methods in the revised Manu highlighted as green color.
- The results comparison with established plant models is missing. The presented highlights (ROS, TCA) are well-known pathways impacted by waterlogging. The authors should focus on novelty and features that were not previously described in model plants.
Response:
Thank you for your propounding comments. We have carefully added more discussion with other plants and try our best to propose our innovation as following.
In summary, using advanced transcriptome sequencing technology combined with physical and biochemical indices, we identified 591 waterlogging response gene resourcing. Moreover, several differentially expressed genes related to waterlogging resistance in P. ostii, which provided new molecular markers and technical support for waterlogging resistance improvement of P. ostii varieties. All these candidate genes provide a strong scientific basis for waterlogging tolerance breeding of P. ostii, helps to screen out waterlogging tolerant varieties, and provides high-quality gense resources for waterlogging resistance breeding and variety improvement of P. ostii in the future. Secondly, the physiological, biochemical and molecular responses of P. ostii to waterlogging stress were explored, which enriched the response mechanism of plants to waterlogging stress and broadened the research scope of waterlogging tolerant crops improvement. Finally, the results of this study provide new ideas for coping with agricultural waterlogging caused by climate change, which has important practical application value. It can help promote the sustainable development of P. ostii industry under complex climate conditions, and also provide a reference framework for the research and improvement of waterlogging resistance of other cash crops.
(Line457-472, page11-12)
Minor issues:
- The introduction contains misleading and incorrect information. H2S is not an oxidizing substance.
Response:
We were really sorry for the careless mistakes. Thank you for your reminder.
The H2S has already been deleted. (Line85, page2)
- The Transcriptomics presentation lacks information on the FC threshold and statistical criteria for selecting significant differences.
Response:
We sincerely appreciate the valuable comments. The FC threshold and statistical criteria for selecting significant differences in Transcriptomics Methods has been added.
Plant roots were collected with two biological replications from the designated time intervals (W0h (T1), W6h (T2), W24h (T3), W72h (T4), W120h (T5), and W168h(T6)) after flooded stress and in the flooded recovery group (WR6h (T7), WR48h (T8) WR96h (T9)). Those samples all frozen in liquid nitrogen. A total of 5 μg RNA per sample was used as input material to generate sequencing libraries using the NEBNext® UltrarNTM RNA Library Prep Kit for llumina® (NEB, USA) following the manufacturer's recommendations. After cluster generation, the libraries were sequenced on an Illumina Hiseg 2500 platform, and paired-end reads were generated.
The sequencing data was filtered with SOAPnuke (v1.5.2), and clean reads were obtained and stored in FASTQ format. The clean reads were mapped to reference full-length transcriptome using HISAT2 (v2.0.4). Bowtie2(v2.2.5) was applied to align the clean reads to the reference coding gene set and then the gene expression levels as FPKM (Fragments Per Kilo basepairs per Million sequencing tags) were calculated using RSEM (v1.2.12). The assembled genes were annotated by being aligned to the NCBI non-redundant nucleotide (Nt) database with BLASTN and NCBI non-redundant nucleotide protein (Nr) database using BLASTX with the option of “-e 1E-5”. A total of 18 samples were measured using an Illumina HiSeq platform, yielding an average of 63.09 M of data per sample (Table S5). The average matching rate of the matched gene set was 59.40%; a total of 115,378 genes were detected (Table S11).
Then the log2-scaled FPKM values of genes were used to calculate the correlation coefficient matrix of samples which was applied to the pheatmap function in the pheatmap library in R to perform hierarchical clustering of samples. The abundances of genes were compared to control to identify Differentially Expressed Genes (DEGs) using the DESeq2 (v1.4.5). Genes with a FDR < 0.05 values and fold change |log2(Fold change)| >1 values were used as the significantly DEGs. Gene Ontology (GO) term and Kyoto Encyclopedia of Genes and Genomes (KEGG) pathway using Phyper in the R package version 4.3.1. The GO terms with multiple test corrected P-values ≤0.05 and KEGG pathways with P-values ≤0.05 were regarded as significantly enriched terms and pathways, respectively. (Line584-612, page15)
Differentially expressed genes (DEGs) (FPKM > 0.5) in at least one compared time point between stress and CK, either waterlogging or waterlogging recovery treatment, were used to construct a network by using WGCNA package (version 1.72). Log2(FPKM+1) values of DEGs were used to calculate the adjacency matrix. The signed gene co-expression and nine physical and biochemical index activity network was constructed by using soft threshold power of 7 and a minimum module size of 18[40]. Modules with cut heights < 0.3 were merged. Gene trait value significance (correlation between gene expression pattern and nine physical and biochemical index activity change) were calculated to rank them. (Line618-625, page15-16)
Figure S1. Compared statistically significant analyses of 13 genes of TCA circle between qPCR results and expressing FPKM value of transcriptome data. Note: T1: T0 h; T2: W6 h; T3: W24 h; T4: W72 h (WR0 h); T5: W120 h; T6: W168 h; T7: WR6 h; T8: WR48 h; T9: WR96 h. FPKM, Fragments Per Kilobase of exon model per Million mapped fragments. Significant differences were calculated using either two-tailed Student’s t test or two-way ANOVA. *P < 0.05, **P < 0.01. (Line346-353, page10)
- "while cysteine and methionine metabolism influence plant superoxide dismutase (SOD) activity" — this information should be supplemented with a reference.
Response:
We would like to thank the reviewer for the constructive comment. We modified this and added a reference.
Acclimate to the low oxygen condition, plants activate the gene encoding enzymes for anaerobic pathways such as Pyruvate decarboxylase (PDC), Alcohol dehydrog-enase (ADH), aldehyde dehydrogenase (ALDH), plant cysteine oxidase (PCO), etc.[29] (Line260-263, page7)
- Shao D, Abubakar A, Chen J, et al. Physiological, molecular, and morphological adjustment to waterlogging stress in ramie and selection of waterlogging-tolerant varieties[J]. Plant physiology and biochemistry, 2024, 216:109101.
(Line745-746, page18)
- The sentence in lines 201-204 does not seem to be well integrated into the manuscript.
Response:
We thank the reviewer for bringing this to our attention. Now it reads, Acclimate to the low oxygen condition, plants activate the gene encoding enzymes for anaerobic pathways such as Pyruvate decarboxylase (PDC), Alcohol dehydrog-enase (ADH), aldehyde dehydrogenase (ALDH), plant cysteine oxidase (PCO), etc.[29] Notably, the glycosphingolipid biosynthesis pathways play a significant role in biofilm structure. In water flooding, the biofilm ensures allowing transport, energy conversion and information transmission to continue. (Line260-266, page7)
- Fig. 4 - Fonts are too small, and gradients for heatmaps are not correctly described in the legend.
Response:
We sincerely appreciate the valuable comments. As suggested by the reviewer, we have modified the font size adjusted from 7pt to 9pt and the figure legends are also described in detail as follows.
Figure 4. Scanning for candidate genes underlying waterlogging stress using Weighted gene co-expression network analysis (WGCNA) and KEGG analysis for RNAseq data.. (A) 18 genes modules revealed by WGCNA analysis. (B) Top 20 KEGG enrichment pathways in turquoise module. (C) Ten KEGG enrichment pathways related waterlogging stress by annotation. (D) 21 genes involved into TCA cycle pathway in turquoise module. W, waterlogging. WR, waterlogging recovery. h, hour. FPKM, Fragments Per Kilobase of exon model per Million mapped fragments.
(Line284-290, page8)
- qPCR validation seems more suitable for supplementary data. It is missing statistical evaluation, and the claim that it is similar to the NGS results is not entirely correct (panels K-M).
Response:
We sincerely appreciate the valuable comments. As suggested, qPCR validation has been treated as supplementary figure1 (Figure S1). Statistically significant analysis has been added, and the claim of qPCR compared with NGS has also revised as follows.
Figure S1. Compared statistically significant analyses of 13 genes of TCA circle between qPCR results and expressing FPKM value of transcriptome data. Note: T1: T0 h; T2: W6 h; T3: W24 h; T4: W72 h (WR0 h); T5: W120 h; T6: W168 h; T7: WR6 h; T8: WR48 h; T9: WR96 h. FPKM, Fragments Per Kilobase of exon model per Million mapped fragments. Significant differences were calculated using either two-tailed Student’s t test or two-way ANOVA. *P < 0.05, **P < 0.01. (Line345-352, page10)
To validate the accuracy of the transcriptome data for P. ostii, and also considerately the important biological significance of TCA circle, 13 candidate point genes involved in the TCA cycle pathway were chosen for quantitative real-time polymerase chain reaction (qRT-PCR) validation. The expression patterns of these 6 genes(MDH、SUCA、LSC1、IDH_1、DLD_2、aceE)according to the qRT-PCR experiments closely similar expression pattern with those observed in the transcriptome data, confirming the reliability of the transcriptome analysis (Figure 6), although the expression of another seven genes showed four (ACLY_1、ACLY_2、DLD_1、IDH_2) showed with tiny diffierence, and three (DLAT、SDH、acnA)with a little differences at few time point (Figure S1). (Line355-366, page10)
- Genes should always be italicized (e.g., L358 - MDH, IDH).
Response:
Thanks for your careful checks. The mistake has been corrected.
Notably, genes such as IDH and MDH, pivotal in the tricarboxylic acid (TCA) cycle, exhibit significant down-regulation under waterlogging stress, with their expression gradually rising during the waterlogging recovery phase[38]. (Line447-489, page112)
- 500 μmol·m2·s-1 is PPFD, not light intensity.
Response:
Thanks. Based on your comments, we have replaced light intensity with PPFD. Such as “the photosynthetic photon flux density was 500 µmol·m2·s-1 “(Line480, page13)
- Kaumas Brilliant Blue G-250 does not exist (Methods).
Response:
Thank you for kindly reminding us. Kaumas Brilliant Blue G-250 Methods has been added into Manu. Now it reads,
The soluble protein was determined using Kaumas Brilliant Blue G-250 (Sharebio, Shanghai, China) as follows. First, 1 ml of crude enzyme solution was pipetted into a test tube, and then 5 ml of Kaumas Brilliant Blue G-250 reagent was added and mixed in thoroughly; the mixture was then left to stand for 2 min. Then, the zero was adjusted with Kaumas Brilliant Blue, and the absorbance value at 595 nm was determined. The calculation formula was as the following equation, Protein content (mg/g) = C x Vt / (W x Vs x 103) x 100%, where C is the soluble protein content (μg) obtained from the standard curve, Vt is the total volume of the extract (ml), Vs is the volume of the aspirated sample liquid (ml); and W is the fresh weight of sample (g).
(Line526-534, page15)
- Supplementary information lists identical titles for Tables S1 and S2. The title is misleading: "Genes involved in the tricarboxylic acid (TCA) cycle pathway according to weighted gene co-expression network analysis" — TCA cycle genes are known and do not require weighted gene co-expression network analysis.
Response:
Thank you for kindly reminding us. Revised as the following:
Table S1: 65 KEGG enrichment pathways analysis with 10,399 genes in cluster 12. Table S2: The expression FPKM value of 81 genes involved into TCA circle in Cluster 12.
(Line580, page16)
- R2# Comments on the Quality of English Language : parts of the manuscript need significant improvement
Response:
Thank you for your careful review of our manuscript. We have conducted a comprehensive language edit and made necessary formatting adjustments to ensure that the manuscript meets the journal's standards.
And, a professional language editing has been done by MDPI author service.

Round 2
Reviewer 2 Report
Comments and Suggestions for Authors
I commend the authors on the many modifications done to the revised manuscript and I appreciate the effort invested.
Issues remaining:
1) The results of root morphology
> The author indicated that WinRHIZO was used to analyze roots. Why are the results of this analysis missing in Figure 1? The claimed percentage in the manuscript is missing SD and the reproducibility and statistical evaluation of these changes is missing.
2) Most of the comments were addressed, but these remained:
- Panel E - Soluble sugar content in roots only rarely exceeds 20% DW. Here, the authors claim 8% FW to be present in the measured samples. That seems to be a significant overestimation and could indicate some issues with the methodology.
Response:
Thank you for your kind reminders. Panel E is revised by adding the results of statistical tests. After carefully checking and reanalyzing the source data again, the results of soluble sugar content were correct (showed in Panel E before, now in Figure 2G).
> I have reviewed the indicated references in response to the review. First, the plots supporting the authors' claim show µg/g, and are thus of three orders of magnitude lower and well within the expected range of free carbohydrate content of plant tissues. The values similar to this study were found in pollen, but that is extremly cabohydrate-rich tissue with relatively low content of water and the value should not be comparable to those foudn in root/shoot tissues per FW.
Please, review again your calculations.
- Panel D - The authors highlight the differences in protein content. However, a two-fold difference in protein content (if statistically significant) only reflects a higher amount of water in waterlogged tissues, and it would be strange to find it substantially higher in WR samples compared to T0h.
Response:
Thank you for your insightful comment and kind suggestion.
A mistake of y scale in Panel D figure has been corrected. The new Panel D picture (Figure 2I now) has been modified as follows..
> I have not found any revision of the protein-related data presented in the revised manuscript.
3) The indicated statistical tests are missing post hoc analysis and results of indicated 2-way ANOVA (interaction between factors) are not reported.
4) Materials & Methods
There are still mistakes that need addressing.
- the equation does not make sense and does not correspond with the listed units in the plot:
Percent soluble sugar (mg/g) was calculated as the following equation, C x Vt (W x Vs x 106) x 100%
- I have not found any evidence that Kaumas Brilliant Blue G-250 exists
Comments on the Quality of English Language
I am a bit baffled by the claim that the manuscript was proofread by a professional service. Just by reading the revised abstract, I can't believe that this could be true. In fact, the new revision seems to have more issues with the language than the original submission.
Author Response
Dear reviewer, thanks for your comments, please see our response in the attachment

Round 3
Reviewer 2 Report
Comments and Suggestions for Authors
1) The results of root morphology
I appreciate the new figure, however, there are some issues with the employed statistics. n.s. depicted in Figure 1F (and in may revised figures) does not make sense and can't be correct. Bars 72 (Waterlogging) and 0 (recovery) seem to be identical. Please, correct or describe correctly in the figure legend.
2) Carbohydrate content
The authors do not acknowledge the mistake, but the old method listed the calculation using FW. However, if these data were normalized to DW (as seems to be the case), I have no reason to question the values. Please, modify the figure legend to indicate which values are being shown (FW or DW).
4) The indicated statistical tests are missing post hoc analysis and results of indicated 2-way ANOVA (interaction between factors) are not reported.
I don't see comments on the factor interaction, nor the standard output of posthoc tests (nor the inidcation which posthoc test was used).
5) Materials & Methods
There are still mistakes that need addressing.
- the equation does not make sense and does not correspond with the listed units in the plot:
The new equation still does not match to the reported units in the figures, tehre are no % in your figures.
6)- I have not found any evidence that Kaumas Brilliant Blue G-250 exists
Response:
We would like to thank the reviewer for the constructive comment. We modified this and added a reference for the methods of soluble protein determined.
> This does not address my issue. The text book listed is not accessible and as it is in Chinesse, it is unlikely that words "Kaumas Brilliant Blue G-250" would be there. Provide CAS number of the chemical used, or correct the name. It is highly probable that this is a misstranslation of Coomassie Brilliant blue G 250
Comments on the Quality of English LanguageI have not found any differences from the previous version of the manuscript, still in need of editing and style/language improvement.
Author Response
Comments 1 :
The results of root morphology
I appreciate the new figure, however, there are some issues with the employed statistics. n.s. depicted in Figure 1F (and in may revised figures) does not make sense and can't be correct. Bars 72 (Waterlogging) and 0 (recovery) seem to be identical. Please, correct or describe correctly in the figure legend.
Response 1: We would like to thank you for your professional and constructive comments, and valuable suggestions. We have updated Figure 1F and this figure legend for clarification. The statistical test between the waterlogging group and waterlogging recovery group had been removed because it did not make sense as reviewer’ suggestions. Figure 2 has also improved.
Comments 2:
Carbohydrate content
The authors do not acknowledge the mistake, but the old method listed the calculation using FW. However, if these data were normalized to DW (as seems to be the case), I have no reason to question the values. Please, modify the figure legend to indicate which values are being shown (FW or DW).
Response 2: Deeply thank you for your professional comments. The mistake has been corrected.
W is the sample dried weight (g). (Line 548 in page 15).
Comments 3:
The indicated statistical tests are missing post hoc analysis and results of indicated 2-way ANOVA (interaction between factors) are not reported.
I don't see comments on the factor interaction, nor the standard output of posthoc tests (nor the inidcation which posthoc test was used).
Response 3: We would like to thank the reviewer for pointing out this issue. The methods of statistical tests have been corrected from “Two-way ANOVA” to “One-way ANOVA” analysis (Line 156 in page 4,Line 209 in page 6, Line 355 in page 11, Line 585 in page 16).
Comments 4:
Materials & Methods
There are still mistakes that need addressing.
- the equation does not make sense and does not correspond with the listed units in the plot:
The new equation still does not match to the reported units in the figures, tehre are no % in your figures.
Response 4: We feel great thanks for your professional review work on our article. The equation has been modified as followings:
(1) Protein content (mg/g) = C x Vt / (W x Vs x 103,( Line 535 in page 15)
(2) Soluble sugar (mg/g) was calculated as the following equation, C x Vt / (W x Vs x 103) (Line 545-546, page 15).
Comments 5:
have not found any evidence that Kaumas Brilliant Blue G-250 exists
Response:
We would like to thank the reviewer for the constructive comment. We modified this and added a reference for the methods of soluble protein determined.
> This does not address my issue. The text book listed is not accessible and as it is in Chinesse, it is unlikely that words "Kaumas Brilliant Blue G-250" would be there. Provide CAS number of the chemical used, or correct the name. It is highly probable that this is a misstranslation of Coomassie Brilliant blue G 250
Response 5:Thank you for pointing out this mistake. The mistake of “Kaumas Brilliant Blue G-250” have been modified to “Coomassie Brilliant Blue solution (G-250)”. (Line 531-532, page 15)

Round 4
Reviewer 2 Report
Comments and Suggestions for Authors
The last revision addressed most of my concerns. Some modifications are not perfect, but many research articles in the field are done similarly, and I don't see any need to further comment on it.
There are still many inconsistencies, and the authors should carefully check the manuscript and correct these. For instance, correlation r = 0.81 with p = 0.078 should not be presented as "significant," as it does not meet the 0.05 threshold.
Comments on the Quality of English LanguageThe language needs significant improvement. There are sentences missing verbs (e.g., 258-261; 275-276). Style and grammar should be checked by an English native speaker.
Author Response
Comments and Suggestions for Authors 1: The last revision addressed most of my concerns. Some modifications are not perfect, but many research articles in the field are done similarly, and I don't see any need to further comment on it. There are still many inconsistencies, and the authors should carefully check the manuscript and correct these. For instance, correlation r = 0.81 with p = 0.078 should not be presented as "significant," as it does not meet the 0.05 threshold. Response 1: We deeply thank all valuable and professional comments that help us to improve the manuscript better. And thank you for pointing out this mistake. I am very sorry for my mistake. p = 0.078 has been modified. Additionally, the black module containing 157 genes was also selected because it had the highest gene traits value (r = 0.81, p = 0.0078) significant correlation with “SOD activity” (Figure 4A). (Line 277-279, page 8 ). Comments on the Quality of English Language 2: The language needs significant improvement. There are sentences missing verbs (e.g., 258-261; 275-276). Style and grammar should be checked by an English native speaker. Response: Thank you for your review. We tried our best to improve the manuscript and made some changes to the manuscript. And a professional MDPI language re-editing has been made and hundreds of revisions to the manuscript have been highlighted in light yellow. 1) About 185 genes in “starch and sucrose metabolism” were highly related to the plant soluble sugar content, while 110 genes in “cysteine and methionine metabolism pathway” influenced plant super oxide dismutase (SOD) activity. (Line 256-259 page 8 ). 2) So, the turquoise module contained 1389 genes that were first selected for exploring candidate co-expression association genes with physiological and biochemical trait changes after waterlogging. (Line 275-277 page 8).
